# In situ monitoring of molecular aggregation using circular dichroism

Haoke Zhang[1,2], Xiaoyan Zheng[3], Ryan T.K. Kwok[2], Jia Wang[4], Nelson L.C. Leung[2], Lin Shi[5], Jing Zhi Sun[1], Zhiyong Tang [5], Jacky W.Y. Lam[2], Anjun Qin [1,4] & Ben Zhong Tang [1,2,4]

The aggregation of molecules plays an important role in determining their function. Electron microscopy and other methods can only characterize the variation of microstructure, but are not capable of monitoring conformational changes. These techniques are also complicated, expensive and time-consuming. Here, we demonstrate a simple method to monitor in-situ and in real-time the conformational change of (R)-1,1′-binaphthyl-based polymers during the aggregation process using circular dichroism. Based on results from molecular dynamics simulations and experimental circular dichroism measurements, polymers with "open" binaphthyl rings are found to show stronger aggregation-annihilated circular dichroism effects, with more negative torsion angles between the two naphthalene rings. In contrast, the polymers with "locked" rings show a more restrained aggregation-annihilated circular dichroism effect, with only a slight change of torsion angle. This work provides an approach to monitor molecular aggregation in a simple, accurate, and efficient way.

[1] MOE Key Laboratory of Macromolecular Synthesis and Functionalization, Department of Polymer Science and Engineering, Zhejiang University, 310027 Hangzhou, China. [2] Department of Chemistry, Hong Kong Branch of Chinese National Engineering Research Center for Tissue Restoration and Reconstruction and Institute for Advanced Study, The Hong Kong University of Science and Technology, Clear Water Bay, Kowloon, Hong Kong, China. [3] Beijing Key Laboratory of Photoelectronic/Electrophotonic Conversion Materials, Key Laboratory of Cluster Science of Ministry of Education, School of Chemistry and Chemical Engineering, Beijing Institute of Technology, 100081 Beijing, China. [4] Center for Aggregation-Induced Emission, SCUT-HKUST Joint Research Institute, State Key Laboratory of Luminescent Materials and Devices, South China University of Technology, 510640 Guangzhou, China. [5] Laboratory for Nanosystem and Hierarchy Fabrication, National Center for Nanoscience and Technology, 100190 Beijing, China. These authors contributed equally: Haoke Zhang, Xiaoyan Zheng. Correspondence and requests for materials should be addressed to A.Q. (email: msqinaj@scut.edu.cn) or to B.Z.T. (email: tangbenz@ust.hk)

The term "chiral" is used in chemistry to describe molecules which are non-superimposable on their mirror image[1–4]. Chiral molecules are abundant in nature, with many natural chemicals such as amino acids and deoxyribonucleic acid (DNA) exhibiting chirality. There are various ways to characterize chirality, and amongst them circular dichroism (CD) is the most commonly used tool to determine the absolute configuration, both in solution and the solid state, because of its simple operation, high accuracy and efficiency[5–8]. Particularly, CD has been widely used to analyze the secondary structure of optically-active biomacromolecules such as proteins and DNA, which is sensitive to environmental variation. The dynamic processes of these chiral molecules are important to their biological activity. For example, the aggregation of proteins is associated with many pathological conditions[9]. How to monitor the conformational changes during protein aggregation is significant to clarify the working mechanism of biological processes. Traditionally, these processes have been studied by techniques such as electron microscopy and NMR spectroscopy. In these methods, pretreatment of samples is required and the analyses are carried out under harsh conditions[10–17]. Thanks to its capability for in-situ and real-time detection, CD spectroscopy may be a good choice to investigate the relationship between molecular aggregation and conformational change. In previous work, the effect of aggregation-induced circular dichroism has been observed in some self-assembly systems[18]. The electron microscopic results suggested that CD-inactive molecules were induced to generate strong CD signals upon helix formation in the aggregate state. However, the detailed conformational changes of a single molecule during the self-assembly process is difficult to determine using CD spectroscopy since the detected CD signal results from the whole assembly structure.

Recently, an opposite effect called aggregation-annihilated circular dichroism (AACD) was observed in 1,1′-binaphthyl derivatives. Further research revealed that the annihilation might be caused by the conformational change of 1,1′-binaphthyl (BN)[19]. BN is a well-studied chiral molecule[20,21]. Mason et al. have set up a classical model to calculate the dependence of the CD couplet intensity ($\Delta\varepsilon_{max}$) and Davydov splitting ($\Delta\lambda_{max}$) on the torsion angle ($\theta$) in 1,1′-binaphthyl[6,22,23]. Figure 1a shows the electronic transitions and coupling between the $^1B_b$ and $^1B_b{}'$ transitions in 1,1′-binaphthyl, while Fig. 1b demonstrates the simulated CD spectra of (R)-1,1′-binaphthol with $\theta$ varying from $-45$ to $-110°$. The plots of $\Delta\varepsilon_{max}/\Delta\varepsilon_{max\text{-}0}$ and ($\Delta\lambda_{max} - \Delta\lambda_{max\text{-}0}$) versus the $\theta$ value are shown in Fig. 1c,d, respectively, where $\Delta\varepsilon_{max\text{-}0}$ and $\Delta\lambda_{max\text{-}0}$ are the values at $\theta = -45°$. To investigate the rate, the first-order derivatives of $d|\Delta\varepsilon_{max}|/d|\theta|$ and $d|\Delta\lambda_{max}|/d|\theta|$ are also plotted in Fig. 1c,d with blue scatter lines. As suggested by Fig. 1, the maximum $|\Delta\varepsilon_{max}|$ is located at around $-65°$ and the $\Delta\lambda_{max}$ gradually decreases when $\theta$ evolves from $-45°$ to $-110°$. The established relationship is expected to determine the absolute configuration of chiral materials using the CD measurement.

In this work, four BN-based chiral polymers are synthesized through simple Suzuki coupling reactions. For the polymers with "open" BN units, the AACD effect is clearly observed. However, the annihilation is restrained once the BN units are "locked". Combining the experimental CD data with the simulation results shown in Fig. 1, the process of molecular aggregation can be monitored simply and accurately. Meanwhile, the relevant molecular dynamics simulation proves that this method is reliable.

## Results

**Synthesis and characterization.** The phenomenon of aggregation-induced emission (AIE) was coined by Tang et al. in 2001. AIE luminogens (AIEgens) such as tetraphenylethylene (TPE) usually possess propeller-shaped structures and show no or weak photoluminescence (PL) when molecularly dissolved in their "good" solvent. However, their emission can be dramatically enhanced once aggregates are formed by adding 'poor' solvent. Thus, AIE exhibits an exact opposite photophysical phenomenon to the traditional aggregation-caused quenching effect discovered by Förster in 1954. Further study has revealed that the restriction of intramolecular motion (RIM) is the cause of the AIE effect. Since TPE possesses attractive PL properties in the aggregate state, it is considered to be a promising candidate to construct chiral materials with unique chiroptical properties[24–26]. The CD signal of binaphthyl and PL signal of TPE can be used as a dual signal to monitor molecular aggregation. In this work, four (R)-1,1′-binaphthyl and TPE-based polymers (P-**1** to P-**4**) were synthesized through simple Suzuki coupling reactions (Fig. 2a). The detailed synthetic procedures are given in Supplementary Note 1. In P-**1** and P-**2**, the TPE unit is linked to an "open" binaphthyl ring at the 6- and 3-position, respectively. P-**3** and P-**4** are structurally similar to P-**1** and P-**2**, respectively but the binaphthyl ring is locked. The structures of all polymers were characterized by $^1H$ and $^{13}C$ NMR spectroscopy with satisfactory results (Supplementary Fig. 1–8). Gel permeation chromatography revealed that their weight-average molecular weight is 11900, 9200, 14000, 4800 and polydispersity is 2.53, 1.87, 2.17, 2.00, respectively (Supplementary Table 1). Thermogravimetric analysis (TGA) revealed that all polymers show 5% weight loss at high temperatures of up to 420 °C (Fig. 2b), suggesting that they are thermally stable.

**Circular dichroism.** According to previous studies, it is anticipated that P-**1** and P-**2** show aggregation-annihilated circular dichroism (AACD) effects, while this effect should be restrained in P-**3** and P-**4** as the 1,1′-binaphthyl moiety is locked by the methylene group. The CD spectra of these polymers in solution and aggregate state were first investigated. All the polymers were molecularly dissolved in tetrahydrofuran (THF). When a poor solvent for the polymer was gradually added in their THF solutions, they started to form aggregates. Then, CD spectra of different polymers at different water fraction ($f_w$) were measured. All the polymers show a strong CD signal in pure THF solution. However, the CD signals of P-**1** and P-**2** at $f_w = 90\%$ are much weaker than those of pure THF solution (Fig. 3a, b). Under the same conditions, the CD signals of P-**3** and P-**4** also become lower at high $f_w$ but the extent is much weaker than in P-**1** and P-**2** (Fig. 3c, d). As expected, both P-**1** and P-**2** show an AACD effect, which suggests that the torsion angle $\theta$ changes during the aggregation process. Once the binaphthyl ring is locked by the methylene group, the rigid structures of P-**3** and P-**4** preserve the CD couplet intensity even in the aggregate state.

The effects of water on the CD couplet molar ellipticity [Θ] and wavelength splitting ($\Delta\lambda_{max}$) have been intensively studied. [Θ] is a parameter to describe the molar ellipticity and is usually obtained from experiments. $\Delta\varepsilon$ is the symbol of molar circular dichroism collected from theoretical calculation. The equation of [Θ] ≈ 3300 $\Delta\varepsilon$ indicates that [Θ] is proportional to the CD couplet intensity $\Delta\varepsilon$. The plot of [Θ]/[Θ]$_0$ against $f_w$ is depicted in Fig. 4a, where [Θ]$_0$ is the molar ellipticity at $f_w = 0\%$ and the values of CD molar ellipticity were acquired from peaks at 280, 240, 280, and 250 nm for P-**1**, P-**2**, P-**3** and P-**4**, respectively. It is clear that the CD signals of P-**1** and P-**2** are tremendously annihilated. Compared to [Θ]$_0$, only 30% of [Θ] remains at $f_w = 90\%$. However, for P-**3** and P-**4**, their [Θ] stays at ~ 80% of the initial value even at $f_w = 90\%$. These results are in good agreement with our hypothesis that the locked structures of P-**3**

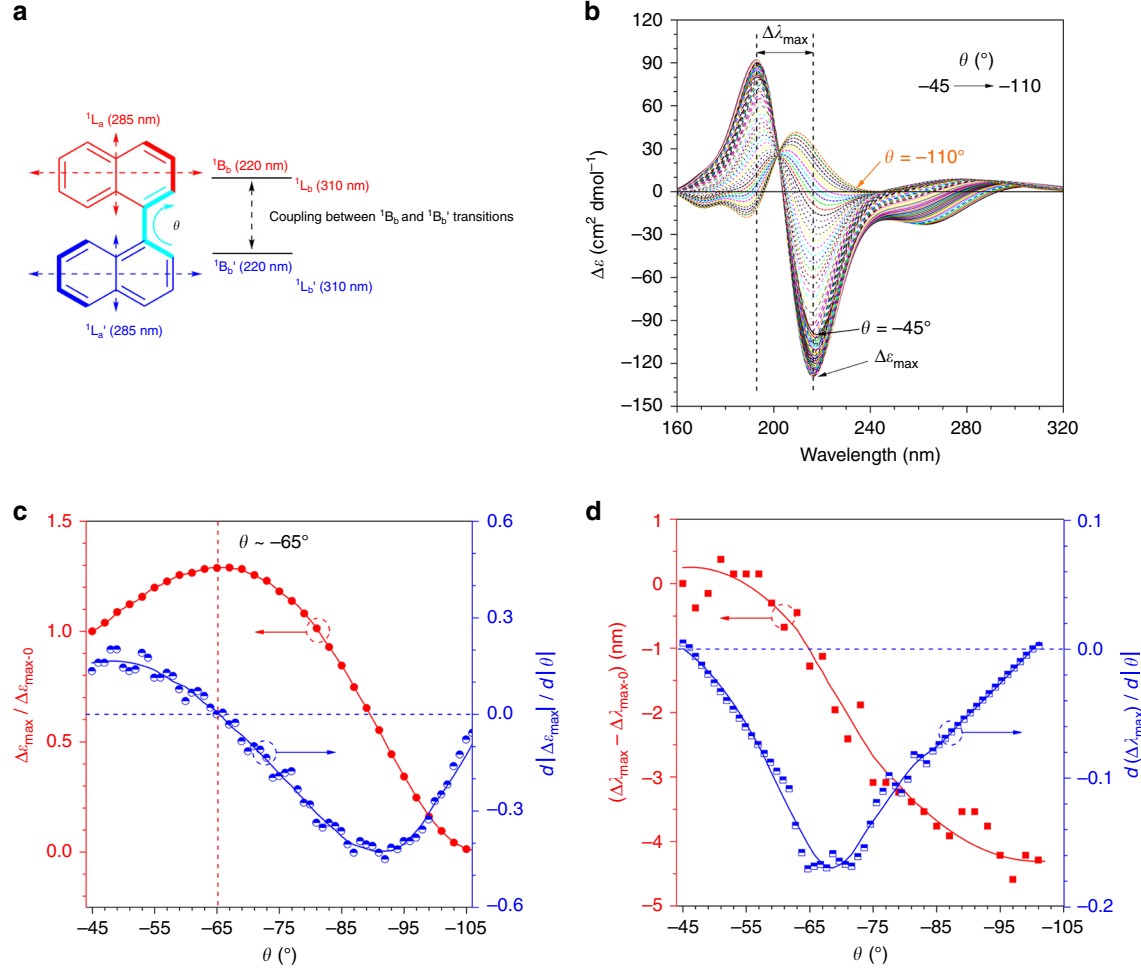

**Fig. 1** Simulated chiroptical properties of binaphthyl. **a** Schematic of polarization directions of the main electronic transition moments and the torsion angle $\theta$ of ($R$)−1,1'-binaphthyl derivatives. **b** Simulated circular dichroism spectra of ($R$)-1,1'-binaphthol with $\theta$ varying from −45 to −110° in 1° intervals. **c** Plot of $\Delta\varepsilon_{max}/\Delta\varepsilon_{max-0}$ versus $\theta$ (red line) and the first-order derivative of $d|\Delta\varepsilon_{max}|/d|\theta|$ (blue line), $\Delta\varepsilon_{max}$ = negative circular dichroism couplet intensity at ~220 nm and $\Delta\varepsilon_{max-0} = \Delta\varepsilon_{max}$ at $\theta = -45°$. **d** Plots of $\Delta\lambda_{max} - \Delta\lambda_{max-0}$ versus $\theta$ (red line) and the first-order derivative of $d|\Delta\lambda_{max}|/d|\theta|$ (blue line). The red dots in Fig. 1c,d were smoothed using the Savitzky–Golay method. $\Delta\lambda_{max}$ = Davydov splitting width shown in Fig. 1b according to the Mason model[23], $\Delta\lambda_{max-0} = \Delta\lambda_{max}$ at $\theta = -45°$. Calculated with CAM-B3LYP/6-31 G*, Nstates = 30, Gaussian 09 Program

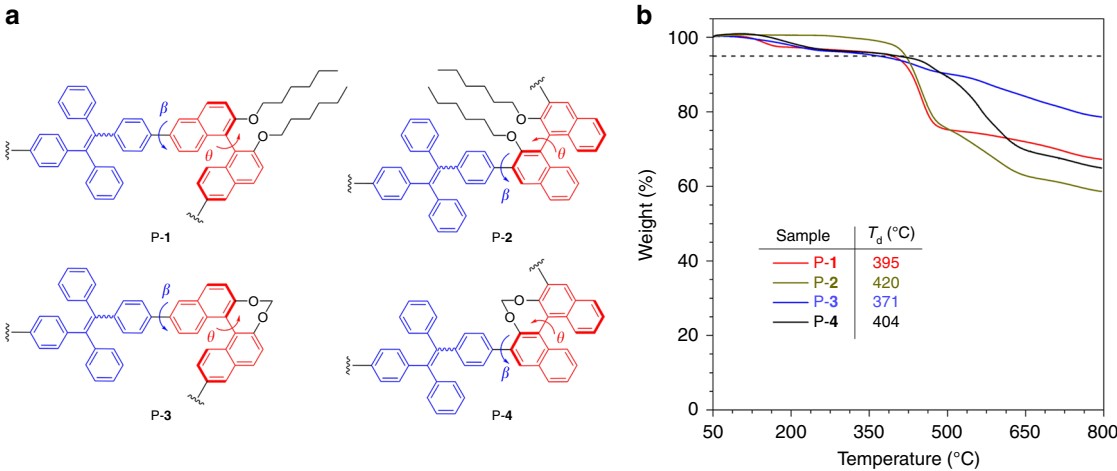

**Fig. 2** Structural and thermodynamic information. **a** Chemical structures of polymers P-**1**, P-**2**, P-**3**, and P-**4**, and their **b** thermogravimetric analysis (TGA) thermograms recorded under $N_2$ at a heating rate of 10 °C/min

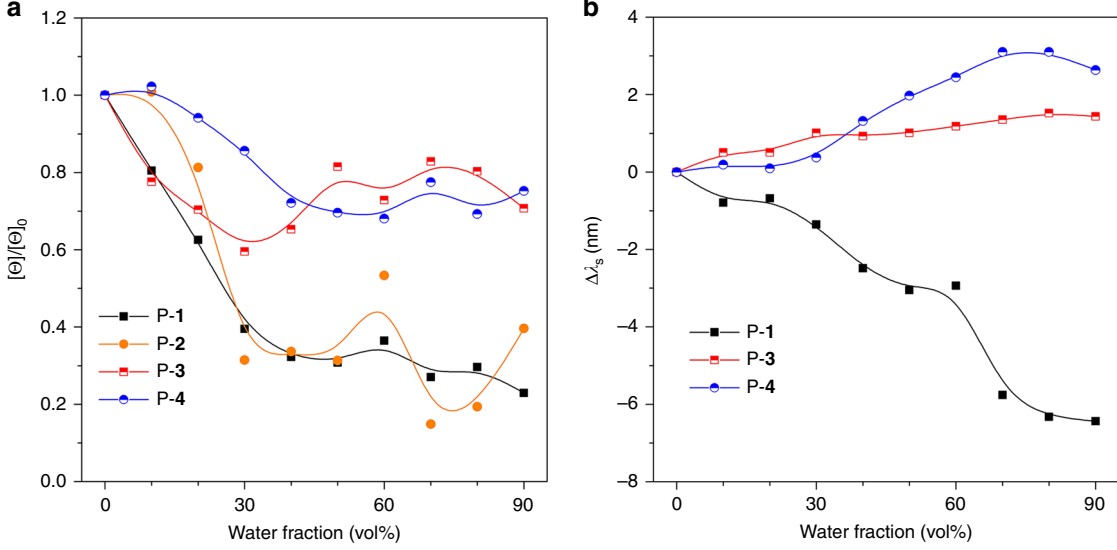

**Fig. 3** Circular dichroism spectra. **a** P-**1**, **b** P-**2**, **c** P-**3**, and **d** P-**4** in THF/water mixtures with different water fractions ($f_w$). Concentration: $10^{-4}$ M

**Fig. 4** Experimental analysis of chiroptical properties within the process of aggregation. **a** Plots of $[\Theta]/[\Theta]_0$ versus the composite of P-**1** to P-**4** in THF/water mixtures at a concentration $10^{-4}$ M. $[\Theta]_0$ = molar ellipticity at water fraction ($f_w$) = 0%. The values of $[\Theta]$ were acquired from the peaks at 280, 240, 280, and 250 nm for P-**1**, P-**2**, P-**3**, and P-**4**, respectively. **b** Plots of $\Delta\lambda_s$ versus $f_w$, where $\Delta\lambda_s = \Delta\lambda_{max} - \Delta\lambda_{max-0}$ and $\Delta\lambda_{max-0}$ = wavelength splitting at $f_w$ = 0%

and P-**4** restrict the rotation of the naphthalene rings to suppress the AACD effect. By careful examination of the CD spectra in Fig. 3, it was found that the wavelength splitting ($\Delta\lambda_{max}$) changes continuously when water is added into the polymer solutions. To make it clear, the plots of $\Delta\lambda_s$ versus $f_w$ are presented in Fig. 4b, where $\Delta\lambda_s = \Delta\lambda_{max} - \Delta\lambda_{max-0}$ and $\Delta\lambda_{max-0}$ = wavelength splitting at $f_w = 0\%$. It was found that the $\Delta\lambda_s$ of P-**1** keeps decreasing during the process of aggregation. In contrast, for the "locked" polymers (P-**3** and P-**4**), their $\Delta\lambda_s$ increases slightly with increasing $f_w$. It is worth mentioning that the positive splitting peak of P-**2** is located in a fairly short-wavelength region, which is even shorter than the cut-off wavelength of THF. Therefore, the $\Delta\lambda_s$ could not be calculated. As the previous study found that the CD couplet intensity and wavelength splitting showed a direct relationship with the torsion angle $\theta$, careful analysis on all these experimental results could provide valuable insights into the conformational change during the aggregation process.

## Discussion

To further study the relationship between CD annihilation and the conformational change during the aggregation process, molecular dynamics (MD) simulation of the polymers was performed in THF and water, respectively. Polymers P-**1** and P-**3** were selected as representatives for the simulation and the details of the calculation are shown in the Methods section. Figure 5a, e show the single P-**1** and P-**3** molecules in THF and simulated aggregates with 30 P-**1** and P-**3** molecules in water are demonstrated in Fig. 5b, f. The MD simulation results indicate that P-**1** shows a broad distribution of dihedral angle $\theta$ but P-**3** exhibits a narrow distribution. For example, the full width at half maximum (FWHM) of P-**1** is 35° and 45° in THF and water, respectively. However, the "locked" polymer P-**3** only possesses a 10° FWHM in both THF and water, suggestive of its higher rigidity. For P-**1**,

the most probable conformation in THF is at $\theta = -76°$. When aggregates are formed in water, the $\theta$ of part of the conformation decreases from $-76°$ to $-102°$ though the most probable conformation is still located at ~75°. Meanwhile, compared with the most probable conformation in THF, only a 2° increase in $\theta$ from $-52°$ to $-50°$ is observed in water for P-**3**. The maintenance of conformation in P-**3** could be attributed to the restriction from the "locked" binaphthyl.

As suggested by the classical model of BN constructed by Mason (Fig. 1), the $\Delta\varepsilon_{max}$ and $\Delta\lambda_{max}$ decrease simultaneously when $\theta$ decreases from $-76°$ to $-102°$. However, when $\theta$ is slightly increased from $-52°$ to $-50°$, the $\Delta\varepsilon_{max}$ decreases but $\Delta\lambda_{max}$ becomes higher. Expectedly, the simulated change of $\Delta\varepsilon_{max}$ and $\Delta\lambda_{max}$ from THF to water coincides with the experimental plots shown in Fig. 4. The conformational change along with molecular aggregation is schematically summarized in Fig. 6. From solution to aggregate, the $\theta$ in "open" polymers (P-**1** and P-**2**) becomes more negative and part of the conformers relax from cisoid to transoid. On the contrary, the $\theta$ in "locked" polymers (P-**3** and P-**4**) increases slightly and the cisoid conformation is preserved throughout the whole aggregation process.

The photophysical properties of the polymers were also studied. The UV spectra of P-**1**, P-**2**, P-**3**, and P-**4** in THF exhibit an absorption maximum ($\lambda_{abs}$) at 347, 333, 337, and 328 nm, respectively. On the other hand, in THF/water mixture (v/v, 7/3), their emission maximum ($\lambda_{em}$) is 506, 498, 499, and 492 nm, respectively (Fig. 7). The photophysical properties and relevant simulation results are summarized in Table 1. From the $\lambda_{abs}$ and $\lambda_{em}$ values, the order of conjugation is: P-**1** > P-**3** > P-**2** > P-**4**, which is well matched with the calculated energy gap (3.186, 3.204, 3.312, and 3.321 eV for P-**1**, P-**3**, P-**2** and P-**4**, respectively).

Study of the $\lambda_{abs}/\lambda_{em}$ change could be another tool to monitor molecular aggregation. TPE is a well-studied molecule with

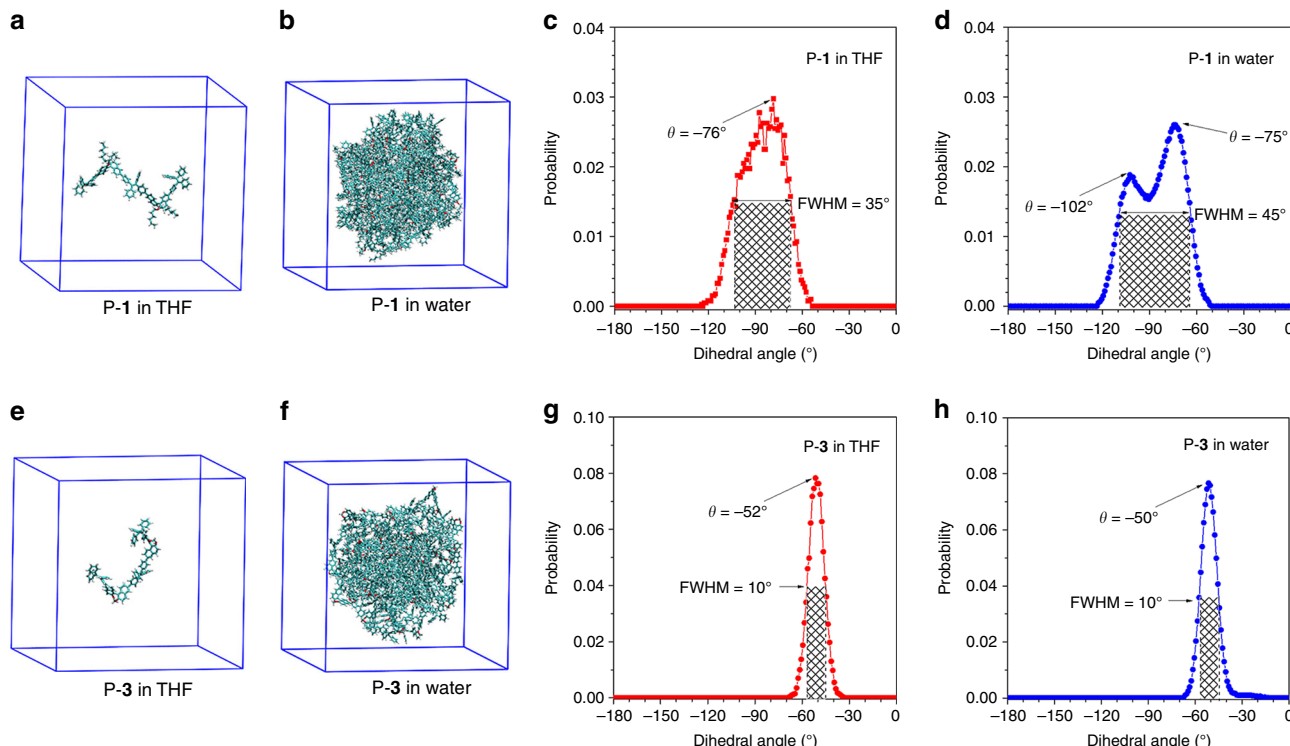

**Fig. 5** The atomistic models of P-**1** and P-**3** in molecular dynamics simulations. **a**, **e** Show single P-**1** and P-**3** molecules in THF solution. **b**, **f** Show 30 P-**1** and P-**3** molecules aggregated in water solution. For clarity, the solvent molecules for THF and water are not shown here. **c**, **d** Show the relative Boltzmann populations of torsion angle $\theta$ in various conformers of P-**1** in THF and water, respectively. **g**, **h** Show the relative Boltzmann populations of $\theta$ in various conformers of P-**3** in THF and water, respectively. FWHM = full width at half maximum

aggregation-induced emission characteristics. Previous studies have shown that the $\lambda_{abs}/\lambda_{em}$ of TPE shifts little when its molecules aggregate. So, the relationship between $\lambda_{abs}$ and $\beta/\theta$ was further investigated by theoretical calculation. A semi empirical CIS/ZINDO method was applied with NStates = 10 and all the calculations were performed using the Gaussian 09 program. The simulation results are shown in Fig. 8 and Supplementary Fig. 10-11. In "open" polymers such as P-**1**, the $\lambda_{abs}$ shows almost no change when $\theta$ decreases from $-80°$ to $-110°$. However, an obvious 5 nm hypochromic shift of $\lambda_{abs}$ is observed when $\beta$ increases from 26° to 46° (Fig. 8a–c). This indicates that the $\lambda_{abs}/\lambda_{em}$ value could be used to monitor the change of $\beta$ in P-**1**. However, in "locked" P-**3**, both $\beta$ and $\theta$ contribute mainly to $\lambda_{abs}$. As shown in Fig. 8d–f, decreasing $\theta$ from $-45°$ to $-55°$ or increasing $\beta$ from 32° to 42° results in a 1 nm hypochromic shift of $\lambda_{abs}$. This suggests that the change of $\lambda_{abs}/\lambda_{em}$ in P-**3** should be ascribed to the variation of $\beta$ and $\theta$. P-**2** shows the same effect as P-**1**, and P-**4** and P-**3** also exhibit similar behaviors.

Then, the PL spectra of these polymers were measured in THF/water mixtures with different $f_w$ (Fig. 9 & Supplementary Fig. 12-13). All the polymers show no or quite weak emission in pure THF solution. However, their emission is tremendously enhanced upon water addition. For example, the PL intensity ($I$) of P-**1** increases by almost 60 times at $f_w = 90\%$, demonstrative of an AIE effect (Fig. 8c)[27–29]. Similarly, the fluorescence quantum yield ($\Phi_F$) of the polymers also increases upon aggregate formation and their maximum solid-state $\Phi_F$ reaches 60% (Supple-

mentary Fig. 14 and Supplementary Table 6). Except for P-**2**, all the polymers show an insignificant change of $\lambda_{em}$ during aggregation (Fig. 9d). For P-**1** and P-**2**, as the decrease of $\theta$ was proved to exhibit negligible impact on $\lambda_{em}$, so we can conclude that the $\beta$ value remains almost unchanged from solution to aggregates in P-**1**. On the other hand, the 14 nm hypsochromic shift of $\lambda_{em}$ in P-**2** suggests an increase of $\beta$ in the aggregate state. In P-**3** and P-**4**, both $\theta$ and $\beta$ play important roles in the $\lambda_{em}$ change. However, the $\lambda_{em}$ stays almost unchanged before and after the aggregation. As suggested by the CD results, only a small increase of $\theta$ is observed in P-**3** and P-**4**. We can infer that the $\beta$ value also suffers small change by aggregation. Among these polymers, it seems that only P-**2** shows an obvious increase of $\beta$ during aggregation presumably due to its crowded structure, and only a slight change of $\beta$ exists in the other three polymers. In order to verify this conclusion, the distribution of $\beta$ for P-**1** and P-**3** obtained from the MD simulation was extracted and the corresponding results are shown in Supplementary Fig. 20-23. Unexpectedly, the dihedral angle $\beta$ is not fixed both in THF and water. There is an interchange between the conformers with $\beta \approx 30°$, $-30°$, $150°$, and $-150°$, which indicate the phenyl ring rotation is active in these polymers but with several probable conformers. From the point of view of electronic conjugation, the energy gap almost shows no difference among $\beta = 30°$, $-30°$, $150°$, and $-150°$. Then all the angles of $\beta$ were transformed and centralized in the range of 0–90°. In this case, the angle with $\beta = 30°$, $-30°$, $150°$ and $-150°$ could be classified into $\beta = 30°$. The processed data are shown in Supplementary Figure 24. From THF to water, the distribution of processed $\beta$ does not show an obvious change both in P-**1** and P-**3**. Meanwhile, their most probable angle is also similar. For example, P-**1** and P-**3** shows the highest probability of $\beta$ at 31° and 32°, respectively, both in THF and water, which is consistent with the PL results. Based on the above analysis, we further conclude that the PL method cannot accurately monitor the conformational change during the process of aggregation, due to the insufficient information provided by the PL spectra. On the contrary, in CD spectra, both molar ellipticity and Davydov splitting could work synergetically to monitor the conformational change, which makes CD spectroscopy a promising tool to monitor molecular aggregation.

In summary, four 1,1′-binaphthyl and TPE-based chiral AIE polymers were designed and synthesized. Polymers P-**1** and P-**2** with "open" binaphthyl exhibited a typical AACD effect and their CD couplet intensity was largely annihilated in the aggregates.

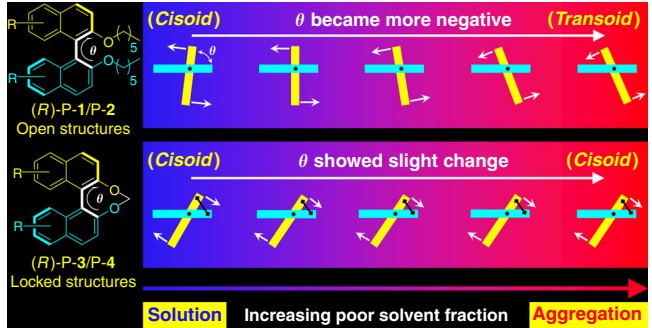

**Fig. 6** Schematic representation of the conformational change of (*R*)−1,1′-binaphthyl moieties during the aggregation process

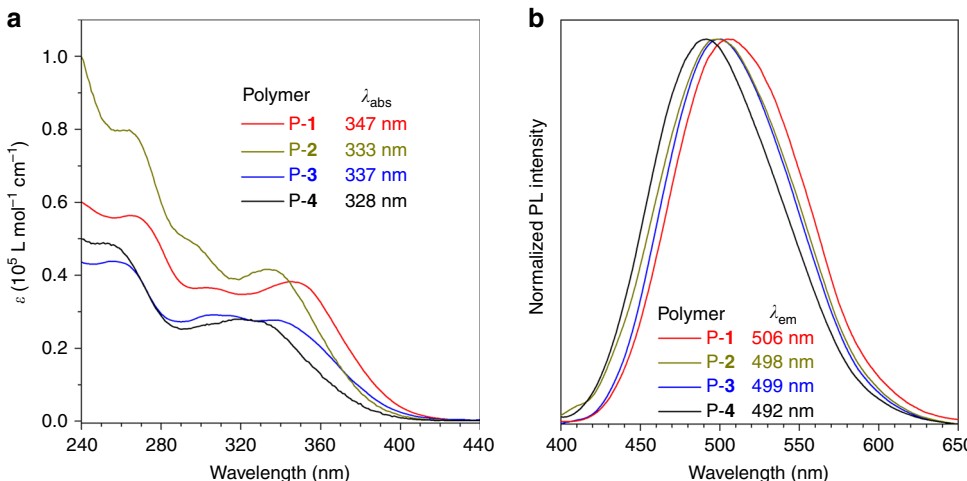

**Fig. 7** Photophysical properties of these four polymers. **a** Ultraviolet absorption spectra of P-**1**, P-**2**, P-**3**, and P-**4** in THF. **b** Normalized photoluminescence (PL) spectra of P-**1**, P-**2**, P-**3**, and P-**4** in THF/water mixture (v/v, 7:3). Concentration: $10^{-5}$ M. $\lambda_{ex}$(nm): 355 (P-**1**), 345 (P-**2**), 345 (P-**3**), and 340 (P-**4**). $\lambda_{abs}$ = absorption maximum, $\lambda_{em}$ = emission maximum

However, the AACD effect was suppressed in P-**3** and P-**4** as the binaphthyl moieties were locked by methylene. These results indicate that the AACD effect is mainly caused by the twisting of naphthalene rings. The combination of MD simulation and analysis on the change of CD couplet intensity and wavelength splitting during the aggregation process is thus an appealing method for in-situ and real-time monitoring of the conformational change. From the solution to aggregate state, the $\theta$ of "open" polymers is largely decreased and part of the conformers

relax from cisoid to transoid. On the contrary, the "locked" polymers show a slightly larger $\theta$ and their cisoid conformation is preserved during the aggregation process. Thus, CD spectroscopy may be used for real-time and in-situ monitoring of molecular aggregation, especially for biomacromolecular aggregations in physiological conditions.

## Methods

**Molecular dynamics simulations.** Molecular dynamics simulations were performed using the GROMACS software package (version 5.1.5)[30]. The atom types and parameters of P-**1** and P-**3** were set from the general amber force field[31]. The electrostatic potential of P-**1** and P-**3** were calculated by Gaussian 09 package[32] and the corresponding partial charges reproducing the electrostatic potential were obtained by the restrained electrostatic potential fit method[33,34]. To mimic the dilute THF solution of P-**1** and P-**3**, the single P-**1** and P-**3** molecules were placed in a cubic box of pre-equilibrated THF molecules[35–37], respectively. For each system, the box length was 6.5 nm containing one P-**1** or P-**3** molecule and 2000 THF molecules (see Fig. 5a, e). To simulate the aggregate state of P-**1** and P-**3**, we first placed 30 P-**1** or P-**3** molecules in a small cubic box with length 4.5 nm. Then we centered the small cubic box in a large cubic box with length 8.0 nm and solvated by the pre-equilibrated TIP3P water molecules (see Fig. 5b, f)[38]. For all the systems, including single molecule THF solution and aggregates of P-**1** and P-**3**, we first carried out energy minimization, followed by 5 ns NPT ($T = 300$ K and $P = 1$ bar) simulations to relax the system. After equilibration, we further ran four independent 50 ns production MD simulations for single P-**1** and P-**3** molecules in THF solution and carried out 6 independent 50 ns production MD simulations for P-**1** and P-**3** aggregates in water solution, respectively (Supplementary Fig. 15–18). The temperature and pressure were controlled by the velocity rescaling thermostat[39] and Berendsen barostat[40] respectively. The time constants of couplings for both temperature and pressure were 0.1 ps. For the electrostatic interactions, the

**Table 1 Photophysical properties and simulation results. Calculated by B3LYP/6-31 G(d), Gaussian 09 program**

| Polymer | $\lambda_{abs}$ (nm) | $\lambda_{em}$ (nm) | $E_g$ (eV) | $\theta$ (°) | $\beta$ (°) |
|---|---|---|---|---|---|
| P-**1** | 347 | 506 | 3.186 | −92.4 | 36.0 |
| P-**2** | 333 | 498 | 3.312 | −78.8 | 44.5 |
| P-**3** | 337 | 499 | 3.204 | −49.0 | 36.7 |
| P-**4** | 328 | 492 | 3.321 | −51.2 | 49.5 |

$\lambda_{abs}$ and $\lambda_{em}$ absorption maximum and emission maximum in THF and THF/water mixture (v/v, 7:3), respectively, $E_g$ calculated energy gap, $\theta$ calculated torsion angle between two naphthalene rings, $\beta$ torsion angle between the naphthalene and phenyl ring of TPE moiety. DFT calculations were carried out to optimize the ground-state conformation of these polymers in the gas phase. To simplify the calculation, only two repeating units were chosen to perform the optimization (Supplementary Fig. 9 and Supplementary Table 2–5)

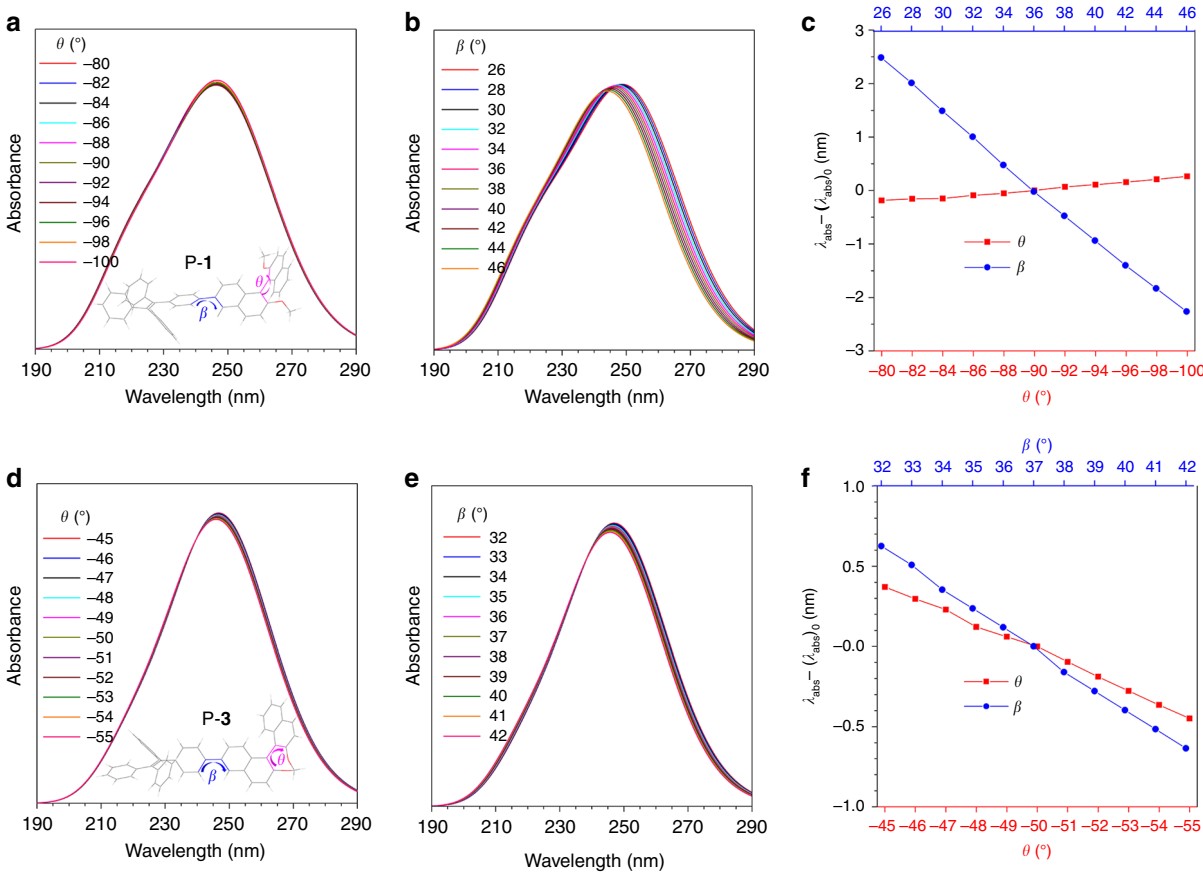

**Fig. 8** Simulated photophysical properties of different conformations. **a**, **b**, **d**, **e** Simulated absorption spectra of **a**, **b** P-**1** and **d**, **e** P-**3** with $\theta$ varying from **a** −80° to −100° and constant $\beta$ (36°) and **d** from −45° to −55° and constant $\beta$ (37°), and $\beta$ varying from **b** 26° to 46° and constant $\theta$ (−90°) and **e** from 32° to 42° and constant $\theta$ (−50°). **c**, **f** Plots of $\lambda_{abs} - (\lambda_{abs})_0$ versus $\theta$ and $\beta$. $\lambda_{abs}$: simulated absorption maximum, $(\lambda_{abs})_0$: simulated absorption maximum wavelength of P-**1** at $\theta = -90°$ and $\beta = 36°$; and P-**3** at $\theta = -50°$ and $\beta = 37°$. $\theta =$ torsion angle between two naphthalene rings, $\beta =$ torsion angle between the naphthalene and phenyl ring of the TPE moiety. A semi empirical CIS/ZINDO method was applied with NStates = 10 and all calculations were performed using Gaussian 09. For the purposes of a clear demonstration and simplified calculation, the hexyl groups are replaced with methyl groups in P-**1**

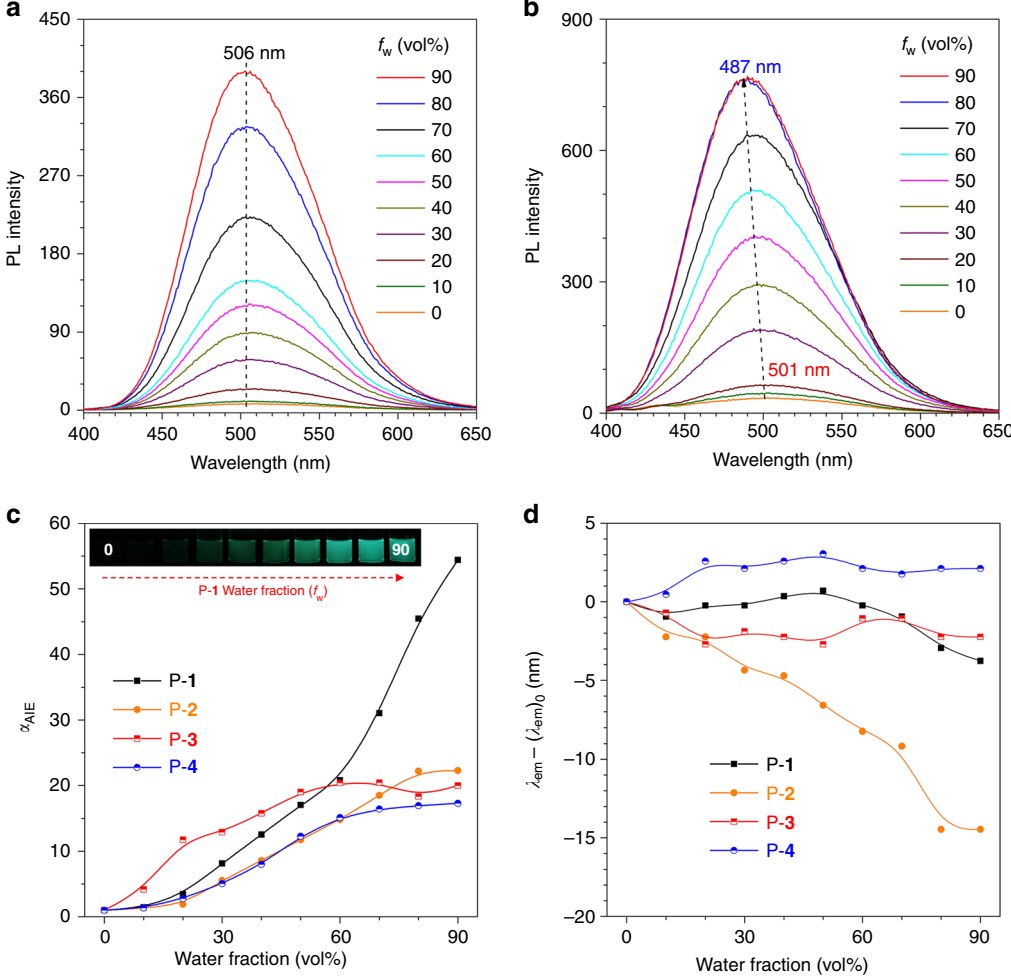

**Fig. 9** Photoluminescence spectra. **a** P-**1** and **b** P-**2** in THF/water mixtures with different water fractions ($f_w$). Concentration: $10^{-5}$ M. $\lambda_{ex} = 355$ nm. **c** Plots of $\alpha_{AIE}$ versus $f_w$, where $\alpha_{AIE} = I/I_0$ and $I_0 =$ maximum emission intensity at $f_w = 0\%$. Inset: photographs of P-**1** in THF/water mixtures with different $f_w$ taken under UV illumination. **d** Plots of $\lambda_{em}-(\lambda_{em})_0$ versus different $f_w$. $\lambda_{em}$: emission maximum of each plot, $(\lambda_{em})_0$: maximum emission wavelength at $f_w = 0\%$

reciprocal space summation was evaluated by the particle mesh Ewald method[41,42]. The direct space summation was computed at a cutoff distance of 1.2 nm. The cutoff distance of VdW interactions was 1.2 nm. All bond lengths here were constrained via the LINCS algorithm[43]. Periodic boundary conditions were applied in all three directions to minimize the edge effects in a finite system. The time step here was 2 fs. The configurations were stored at a time interval of 40 ps for statistical analysis of dihedral angles and the corresponding free energy calculation. In total, for single molecule THF solution, we collected 25,000 configurations, while for aggregates, we collected 37,500 conformations. The obtained distributions of the dihedral angle and the Gibbs free energy were demonstrated in Fig. 5 and Supplementary Fig. 19, respectively.

**Quantum mechanics calculations**. All the quantum mechanism calculations here were performed by the Gaussian 09 program. The optimized conformations of these four polymers were calculated in the gas phase by B3LYP/6-31 G(d). To simplify the calculation, only two repeating units were chosen to do the optimization. Calculation on the CD spectra of BINOL was carried out at CAM-B3LYP/6-31 G* level with Nstates = 30. Calculation on the UV spectra of the polymers was performed with semiempirical CIS/ZINDO method with NStates = 10. The CD and UV spectra were processed by Multiwfn 3.6 Program[44].

## Data availability

The authors declare that the data supporting the findings of this study are available within the article and its Supplementary Information files. Additional data are available from the corresponding author upon reasonable request.

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

## Acknowledgements

This work was partially supported by the National Natural Science Foundation of China (21788102, 21490571, and 21803007), the National Program for Support of Top-Notch Young Professionals, the Innovation and Technology Commission of Hong Kong (ITC-CNERC14SC01); the Research Grants Council of Hong Kong (16308016, C2014-15G, C6009-17G and A-HKUST605/16), and the Technology Plan of Shenzhen (JCYJ20160229205601482).

## Author contributions

H.Z., A.Q., and B.T. designed the experiments. H.Z. synthesized polymers and measured all the PL, UV, GPC and NMR spectra. H.Z. and X.Z. carried out the theoretical calculations and results analyses. H.Z., R.T.K.K., N.L.C.L., and J.S. discussed the molecular dynamics. H.Z., J.W., L.S., and Z.T. characterized and analyzed the chiroptical properties. H.Z. obtained the quantum yield measurements. H.Z., R.T.K.K., and J.W.Y.L. revised the manuscript. H.Z., A.Q., and B.T. wrote the manuscript with comments from all authors.

## Additional information

**Competing interests:** The authors declare no competing interests.

