## [Peer Review File · Nature Communications]

Reviewer #1 (Remarks to the Author):

The manuscript entitled "Monitoring Molecular Dynamics Using Circular Dichroism" by Dr. Tang and coworkers represents an interesting and application-oriented investigation of four 1,1-dinaphthyl and tetraphenylethylene based chiral polymers, demonstrating that CD spectroscopy can be used as a real-time sensor of macromolecular aggregations. Such spectroscopic outcome is important for sensing aggregation of biological molecules often linked to neurodegenerative disorders. From application-perspective, the presented research is of high impact.

Authors provide meticulously collected experimental CD-based evidence that "opened" binaphthyl polymers P-1 and P-2 exhibit prominent aggregation-anihilated circular dichroism AACD effect with down-modulated intensity upon aggregation. On the other hand, the "locked" polymer-analogs P-3 and P-4 exhibit a more restrained AACD effect as a result of aggregation. Overall, AACD effect has been attributed to twist of the naphthalene rings (more negative for P-1 and P-2 vs. less negative for P-3 and P-4), which is most graphically clearly summarized in Figure 6.

Major revisions:

The manuscript is suggested for publication upon major revision which will make CD analysis more comprehensive and increase the confidence level for the proposed link between suggested conformational changes and observed diagnostic CD changes upon aggregation. Specifically, it is suggested with this review that authors predict and simulate theoretical CD signal for proposed optimized model-geometries of solution and aggregation state conformations, as provided for truncated models given in Figure 5. Such calculations could be achieved with ZINDO and/or CAM-B3LYP/6-31G* levels of theory for ~30 electronic transitions. Subsequently, the simulated CD responses can be correlated with experimental data in order to provide more context and higher confidence level towards the interpretation of the experimentally obtained diagnostic CD signatures.

Additionally, since empirical studies depend on solvent fraction (fw) the same solvation effect which leads to aggregation should be emulated in molecular modeling efforts. For example, authors should predict Gibbs free energies and relative Boltzmann populations of various conformations and subsequently demonstrate higher relative stabilities of conformations given in Figure 5 vs. those anticipated as less favorable for given solvent medial (aggregation extent).

Minor corrections:

In Introduction paragraph (line 32), grammar should be correct from “rare” to “rarely” in the first sentence. In general, throughout the manuscript sentences should be simplified/improved for better flow and grammar should be double-checked.

Resolution of θ designations in Figure 1, given on the right in the blue text should be improved.

In Figure 8 y-axis label α_{AIE} and in the conclusion paragraph, the acronym “AIE”, should be explained somewhere in the earlier text.

Reviewer #2 (Remarks to the Author):

The paper reports that easy, accurate and efficient way to monitor the conformational change of the 1,1'-binaphthyl based polymer induced by aggregation of the molecules. The authors proposed that the torsion angle of binaphthyl can be easily determined by CD spectroscopy using the relationship between $\Delta\varepsilon/\Delta\lambda_{max}$ and torsion angle of the naphthalene units. The author also done the DFT calculations to interpret the experimental results. I think that the results are well organized and discussion is quite convincing; thus, the study provides an important contribution to the field of chemistry, especially structural organic chemistry field. Consequently, I would like to recommend that the manuscript should be accepted for publication in Nature Commun. Before publication, the authors should reconsider and revise the manuscript about several minor points listed below.

1. I feel that the word of “molecular dynamics” in the title gives misleading to the readers; because the proposed method can provide only an information about the static or average conformation of molecules in the solution and aggregates. I think that the word of “molecular dynamics” is associated with the dynamic change of the conformation, namely the process of the change from one conformation to the other conformation, induced by stimuli such as addition of the poor solvent. Thus, I recommend the authors to reconsider the title.
2. Because many readers of this journal is not an expert in CD spectroscopy, the author should explain the relationship between $\Delta\varepsilon$ and $[\theta]$. They describe it briefly in p. 5 in the manuscript; however, I think it is not enough to understand their relationship.

3. In fig 5, the torsion angle between two naphthalene unit is expressed with a symbol α . On the other hand, in fig 6 and main text, it is expressed with θ . It is very confusing to the readers, and the symbol must be consistent.

4. Are the aggregates of those polymers crystal? Is there any evidence if the aggregates are crystal or amorphous? If the aggregates are amorphous, the angle θ has some distribution. Is there any effect of the distribution of the θ angle on the $\Delta\varepsilon$, $[\Theta]$, and $\Delta\lambda_{\max}$?

Reviewer #3 (Remarks to the Author):

The modulation of the CD signal of binaphthalene when conformational changes occur is known in the literature, but here the authors use CD and photoluminescence (PL) to evaluate conformational changes upon aggregation of polymers based on (R)-1,1'-binaphthyl (BN) and tetraphenylene (TPE). Four polymers were studied (P1 to P4), depending on the location of the bond between BN and TPE, and whether BN was locked or not.

This paper needs major revisions.

First, the title must be changed: "Monitoring molecular dynamics using circular dichroism". It is abusive to use the terms "molecular dynamics" of aggregation. What is observed is changes of CD and PL spectra upon adding water to a THF solution of the polymer, which causes aggregation. Then, using models relating CD and PL spectra to the torsions in the polymers, the authors try to evaluate whether the torsions change upon aggregation and in which direction (larger or smaller angles). What the authors intend to do is thus to monitor aggregation and conformational changes resulting from that aggregation, not the dynamics of the process.

In the polymers, there are two torsions that critically affect the conformations, and the authors aim to evaluate how these torsions change upon aggregation:

- 1) the torsion β about the bond linking tetraphenylene (TPE) and binaphthyl (BN);
- 2) the torsion θ within BN

β is studied by PL experiments and DFT simulations. Based on energy gaps calculated by DFT, the authors estimate that $\lambda(\text{abs})$ (PL) that is lowest in energy mainly depends on β but not θ . As $\lambda(\text{abs})$ decreases in the following order: P1 - P3 - P2 - P4 for polymers in THF, and as β (calculated by DFT) increases following the same order, the authors conclude that $\lambda(\text{abs})$ could be used to monitor β .

I think that their conclusion is not supported by the data. First, the authors did not take count in their analysis that TPE and BN are not grafted at the same positions in the four polymers, and this aspect alone can change dramatically the absorption spectra. P1 and P3 have TPE and BN grafted at the same position; P2 and P4 are grafted at another position. I think that this aspect alone explains why P1-P3 have higher $\lambda(\text{abs})$ than P2-P4. Then, within the group P1-P3, $\lambda(\text{abs})$ differs by 10 nm between P1 and P3, but β in the two polymers differs by only 0.7°. How can the authors thus claim that β mainly explains the shift of the absorption band in these polymers? Thus, all the analyses written from line 161 to 202, in the paragraph titled "Photophysical properties and aggregation-induced emission", must be deeply revised if any valuable information is sought. For instance, the influence of structural parameters other than β on the absorption can be obtained by performing simulations where β is fixed in the four polymers; absorption spectra can be generated from excited states obtained from CIS calculations performed on isolated molecules using the ZINDO parameterization.

The δ torsion is studied by CD experiments coupled to the Mason's model, which calculates how the CD signal intensity ($\Delta\epsilon$ or $[\Theta]$) and Davydov splitting ($\Delta\lambda$) evolve with θ . Based on these methods, the authors claim that upon aggregation, P1 has θ evolving from -92° (isolated chain calculated by DFT) towards -110°, as both the CD intensity and $\Delta\lambda$ simultaneously decrease in the Mason model when the angle evolves from -92° to -110° (at 110°, the signal and $\Delta\lambda$ are null), and that such evolution is also observed in CD experiments. However, when increasing the water fraction, the drops of the CD intensity and of $\Delta\lambda$ look successive, not simultaneous. Up to 30% water, the CD signal intensity drops dramatically, then stabilizes. For $\Delta\lambda$, it is the opposite; up to 30% water, $\Delta\lambda$ drops by only ~1.5 nm, and after 30% water, there is another loss of 4.5 nm. How can the authors rationalize these observations? In any case, to better compare these simulation and experimental results, it should be nice to use identical graphical representation. Experimentally, $[\Theta]/[\Theta]_0$ and $\Delta\lambda - \Delta\lambda_0$ are displayed, where 0 refers to the molecule in pure THF; it should be nice to show the Mason model with $\Delta\epsilon/\Delta\epsilon_0$ and $\Delta\lambda - \Delta\lambda_0$, and not $\Delta\lambda$ and $\Delta\epsilon$ only, with $\Delta\lambda_0$ and $\Delta\epsilon_0$ corresponding to $\theta = -92^\circ$ (for P1).

In addition, I think there are errors in the graphics of Figure 4, as these graphics are not totally consistent with the graphics of Figure 3. For P1, the dots at 80 and 90% water must probably be permuted, as the color coding in Figure 3 indicates that the strongest attenuation of intensity is for 90% water; for P2, at 10% water, the signal is attenuated by 10% while in Figure 3 the signal has the same intensity as for 0% of water; also for P2, for 60% water, the signal is attenuated by 80%, while in Figure 3, the attenuation looks more close to 50%.

Response Letter to the Reviewers

Dear reviewers,

On behalf of all the contributing authors, I really appreciate the valuable comments from you, which are useful to improve the quality of our manuscript. The reviewers' comments are laid out below in italicized and underlined font and specific concerns have been numbered. Our response is given in normal font and the changes/additions are given in red color in the revised manuscript.

Definition of figure:

Figure Rxx: Figures to response the comments of reviewers.

Figure Sxx: Figures given in the supporting letter.

Figure xx: Figures in the main text of this manuscript.

Reviewer 1:

The manuscript entitled "Monitoring Molecular Dynamics Using Circular Dichroism" by Dr. Tang and coworkers represents an interesting and application-oriented investigation of four 1,1-dinaphthyl and tetraphenylethylene based chiral polymers, demonstrating that CD spectroscopy can be used as a real-time sensor of macromolecular aggregations. Such spectroscopic outcome is important for sensing aggregation of biological molecules often linked to neurodegenerative disorders. From application-perspective, the presented research is of high impact.

Firstly, I really appreciate such a positive feedback from our reviewer. I would like to say that the comments and questions raised by the reviewer do help a lot to improve the quality of our manuscript. I have carefully read each of the comments and tried our best to clarify them through experiments and theoretical calculations. We hope the reviewer satisfied our answers.

Authors provide meticulously collected experimental CD-based evidence that "opened" binaphthyl polymers P-1 and P-2 exhibit prominent aggregation-anihilated circular dichroism AACD effect with down-modulated intensity upon aggregation. On the other hand, the "locked" polymer-analogs P-3 and P-4 exhibit a more restrained AACD effect as a result of aggregation. Overall, AACD effect has been attributed to twist of the naphthalene rings (more negative for P-1 and P-2 vs. less negative for P-3 and P-4), which is most graphically clearly summarized in Figure 6.

Major revisions:

- 1) The manuscript is suggested for publication upon major revision which will make CD analysis more comprehensive and increase the confidence level for the proposed link between suggested conformational changes and observed diagnostic CD changes upon aggregation. Specifically, it is suggested with this review that authors predict and simulate theoretical CD signal for proposed optimized model-geometries of solution and aggregation state conformations, as provided for truncated models given in Figure 5. Such calculations could be achieved with ZINDO and/or CAM-B3LYP/6-31G* levels of theory for ~30 electronic transitions. Subsequently, the simulated CD responses can be correlated with experimental data in order to provide more context and higher confidence level towards the interpretation of the experimentally obtained diagnostic CD signatures.

Thanks for the useful suggestions from our reviewer. Following the comments, circular dichroism (CD) spectra of P-1 to P-4 at their optimized conformation in the gas phase were simulated (shown in Figure S9). The simulated spectra were consistent with the experimental ones except for P-2 (Figure R1). To verify the AACD effect, molecular dynamics calculation was carried out in THF and water solution, respectively. The representative polymer P-1 and P-3 with two repeating units were truncated to setup molecular dynamics simulations. Details of the calculation were shown below.

Figure S9. Optimized structures of P-1, P-2, P-3 and P-4 in the ground state calculated by the DFT at the B3LYP/6-31G(d) level. Only one repeating unit was drawn for clear presentation.

Figure R1. Simulated circular dichroism (CD) spectra of A) P-1, B) P-2, C) P-3 and D) P-4 at their local minimum in the gas phase, Calculated by CAM-B3LYP/6-31G*, Nstates = 30.

All the molecular dynamics (MD) simulations were performed using the GROMACS software package (version 5.1.5). To mimicked the dilute THF solution of P-1 and P-3, the single P-1 and P-3 molecule were placed in a cubic box of pre-equilibrated THF molecules, respectively. For each system, the box length is 6.5 nm, containing one P-1 or P-3 molecule and 2000 THF molecules (see Figure 5A & E). The aggregated state of P-1 and P-3 were obtained by placed 30 P-1 or P-3 molecules in a cubic box with length 8.0 nm and solvated by the pre-equilibrated TIP3P water molecules (see Figure 5B & F). For single molecule THF solution of P-1 and P-3, we run 4 independent 50 ns production MD simulations. For the nano-aggregates of P-1 and P-3, we carried out 6 independent 50 ns production MD simulations for P-1 and P-3 aggregates in water solution, respectively. The time step here is 2 fs. The configurations were stored at a time interval of 40 ps for dihedral angles statistical analysis and the corresponding free energy calculation (Figure S15-19). In total, for single

molecule THF solution, we collected 25000 configurations, while for aggregates, we collected 37500 conformations.

Figure 5. The atomistic models of P-1 and P-3 in molecular dynamics simulations. A) and E) showed single P-1 and P-3 molecule in THF solution. B) and F) demonstrated thirty P-1 and P-3 molecules aggregated in water solution. For clarity, the solvent molecules THF and water were not shown here. C) and D) showed the relative Boltzmann populations of various conformers of P-1 in THF and water, respectively. G) and H) showed the relative Boltzmann populations of various conformers of P-3 in THF and water, respectively. FWHM: full width at half maximum.

As shown in Figure 5, P-1 showed a broad distribution of dihedral angle θ but P-3 exhibited a narrow one. For P-1, the most probable conformation in THF was at $\theta = -76^\circ$. However, another local probable conformation was observed at $\theta = -102^\circ$ in water. This should be the cause of the AACD effect in P-1. Meanwhile, compared with the most probable conformation in THF, only a 2° increase in θ was observed in water for P-3. The stable conformation of P-3 should be attributed to the restriction of the “locked” binaphthyl from twisting. The results from molecular dynamics simulation were very much as we expected. Since an obvious conformational change was observed in P-1, change of its CD spectrum at different θ was also investigated and the results were shown in Figure R2. Unexpectedly, the simulated CD spectra of the present polymers deviated from the reported data. For example, in P-1, the calculated $\Delta\epsilon$ at around 230 nm continuously increased when the dihedral angle θ evolved from -45 to -125° . However, as shown in Figure 1C, the $\Delta\epsilon_{\max}$ of (*R*)-1,1'-binaphthol (BINOL) increased from -45 to -65° , but decrease afterward.

Figure R2. Simulated circular dichroism spectra of A) P-1, B) P-2, C) P-3 and D) P-4 at different conformations, calculated by CAM-B3LYP/6-31G*, Nstates = 30, Gaussian 09 Program.

Figure 1. A) Schematic of polarization directions of the main electronic transition moments and the torsion angle (θ) of (*R*)-1,1'-binaphthyl derivatives. B) Simulated circular dichroism spectra of (*R*)-1,1'-binaphthol with θ varying from -45 to -110° of 1° interval. C) Plot of $\Delta\epsilon_{\max}/\Delta\epsilon_{\max-0}$ versus θ (red line) and the first-order derivative of $d|\Delta\epsilon_{\max}|/d|\theta|$ (blue line), $\Delta\epsilon_{\max}$ = negative CD couplet intensity at around 220 nm and $\Delta\epsilon_{\max-0} = \Delta\epsilon_{\max}$ at $\theta = -45^\circ$. D) Plots of $\Delta\lambda_{\max} - \Delta\lambda_{\max-0}$ versus θ (red line) and the first-order derivative of $d|\Delta\lambda_{\max}|/d|\theta|$ (blue line). $\Delta\lambda_{\max}$ = Davydov splitting width shown in Figure 1B according to the Mason model²³, $\Delta\lambda_{\max-0} = \Delta\lambda_{\max}$ at $\theta = -45^\circ$. Calculated by CAM-B3LYP/6-31G*, Nstates = 30, Gaussian 09 Program.

Since the relationship between $\Delta\epsilon_{\max}$ and θ has already been reported and verified by many groups and our simulation results on BINOL was consistent with the literature (Figure 1), we proposed that the abnormal simulation results of the present polymers were due to the presence of tetraphenylethylene unit (TPE). Then, the simulated calculation was also carried out for TPE and the results were shown in Figure R3. Indeed, a pair of opposite CD spectra were obtained from (*P*)-TPE and (*M*)-TPE. Actually, the chirality of TPE has already been proved by several groups (Zhang et al. *J. Phys. Chem. C*, 2017, 121 (38), 20947; Zheng et al. *J. Am. Chem. Soc.*, 2016, 138 (36), 11469; Cao et al. *J. Am. Chem. Soc.*, 2017, 139 (50), 1814). Experimentally, the chirality of TPE was

only observed in the crystal state or its locked configuration. In general, racemization was found in its solution and aggregate state. Unlike experiment, the CD spectrum simulation was usually carried out for a particular conformer of TPE (*P* or *M*). So, the TPE moiety contributed a lot to the simulated CD spectra of these polymers.

Figure R3. Simulated circular dichroism spectra of (*P*)-TPE and (*M*)-TPE calculated by CAM-B3LYP/6-31G*, Nstates = 30, Gaussian 09 Program.

In summary, the simulated CD spectra of the polymers at different torsion angle θ were influenced by the achiral TPE moiety and the obtained results did not match with the experimental data. However, the molecular dynamics calculation supported the decrease of θ in “open” polymers from solution to aggregates and a slight θ increase in “locked” polymers under the same condition. At last, the establishment of the relationship between $\Delta\epsilon_{\max}$ and θ for BINOL provided a good explanation on the AACD effect.

- 2) Additionally, since empirical studies depend on solvent fraction (f_w) the same solvation effect which leads to aggregation should be emulated in molecular modeling efforts. For example, authors should predict Gibbs free energies and relative Boltzmann populations of various conformations and subsequently demonstrate higher relative stabilities of conformations given in Figure 5 vs. those anticipated as less favorable for given solvent medial (aggregation extent).

Following the review’s comments, molecular dynamics (MD) simulations of these polymers were carried out in THF and water solutions. P-1 and P-3 with two repeating units were chosen to setup the molecular dynamics simulations. Details of the MD simulation were shown in the response of the first question. Four and six independent trajectories were run in THF and water, respectively. These original results were shown in Figure S15-18.

Figure S15. Distribution of dihedral angle θ of P-1 in THF solution. A) to D) Representing four independent MD trajectories.

Figure S16. Distribution of dihedral angle θ of P-1 aggregate in water solution. A) to F) Representing six independent MD trajectories.

Figure S17. Distribution of dihedral angle θ of P-3 in THF solution. A) to D) Representing four independent MD trajectories.

Figure S18. Distribution of dihedral angle θ of P-3 aggregate in water solution. A) to F) Representing six independent MD trajectories.

Then, the average results of the distribution of θ were summarized in Figure R4. P-1 showed a broad distribution of dihedral angle θ while P-3 exhibited a narrow one. For P-1, the most probable conformation in THF was at $\theta = -76^\circ$. However, another local probable conformation was observed at $\theta = -102^\circ$ in water and the most probable conformation was kept at $\theta = -76^\circ$. Compared with the most probable conformation in THF, only a 2° increase of θ was observed in water for P-3. The stable conformation in P-3 could be attributed to the restriction of the “locked” binaphthyl from twisting. The plots of Gibbs free energy were demonstrated in Figure S19. For P-1, it seems that the optimized conformation from quantum mechanics (QM) calculation was slightly different from molecular dynamics simulation. The DFT calculation suggested θ of -92° in gas phase but the most probable conformation in THF was at $\theta = -76^\circ$ according to the MD simulation. In P-3, QM and MD calculations gave a quite close θ value. In any case, the obtained distributions of the dihedral angle and the Gibbs free energy demonstrated that the favorable conformations of P-1 in THF and water were different. In THF, most of the conformers existed at $\theta = -76^\circ$. When aggregates formed in water, part of the conformation decreased their θ from -76° to -102° even the most probable conformation was still located at around 75° . That was the reason why AACD effect was observed in P-1 and P-2. Meanwhile, the negligible increase of θ in P-3 kept a high $[\Theta]$ in the aggregate state instead of annihilation.

Figure R4. The average Boltzmann populations of various conformations for (A and B) P-1 in THF and water, and (C and D) P-3 in THF and water.

Figure S19. The calculated Gibbs free energies of various conformations of (A and B) P-1 in THF and water. (C and D) P-3 in THF and water.

Minor corrections:

- 3) *In Introduction paragraph (line 32), grammar should be correct from “rare” to “rarely” in the first sentence. In general, throughout the manuscript sentences should be simplified/improved for better flow and grammar should be double-checked.*

Sorry for these grammatical mistakes, we have corrected it in our main text. According to the reviewer’s suggestions, we have further polished our manuscript with the help of several English native speakers.

- 4) *Resolution of θ designations in Figure 1, given on the right in the blue text should be improved.*

We have redrawn Figure 1 and the new figure now shows a high resolution.

Figure 1. A) Schematic of polarization directions of the main electronic transition moments and the torsion angle (θ) of (*R*)-1,1'-binaphthyl derivatives. B) Simulated circular dichroism spectra of (*R*)-1,1'-binaphthol with θ varying from -45 to -110° of 1° interval. C) Plot of $\Delta\epsilon_{\max}/\Delta\epsilon_{\max-0}$ versus θ (red line) and the first-order derivative of $d|\Delta\epsilon_{\max}|/d|\theta|$ (blue line), $\Delta\epsilon_{\max}$ = negative CD couplet intensity at around 220 nm and $\Delta\epsilon_{\max-0} = \Delta\epsilon_{\max}$ at $\theta = -45^\circ$. D) Plots of $\Delta\lambda_{\max} - \Delta\lambda_{\max-0}$ versus θ (red line) and the first-order derivative of $d|\Delta\lambda_{\max}|/d|\theta|$ (blue line). $\Delta\lambda_{\max}$ = Davydov splitting width shown in Figure 1B according to the Mason model²³, $\Delta\lambda_{\max-0} = \Delta\lambda_{\max}$ at $\theta = -45^\circ$. Calculated by CAM-B3LYP/6-31G*, Nstates = 30, Gaussian 09 Program.

5) In Figure 8 y-axis label α_{AIE} and in the conclusion paragraph, the acronym “AIE”, should be explained somewhere in the earlier text.

We add the description of “AIE” at the beginning of manuscript when we introduced tetraphenylethylene (TPE) because TPE was a typical luminogen with AIE effect. The description was presented in this way.

“The phenomenon of aggregation-induced emission (AIE) was coined by Tang et al. in 2001. AIE luminogens (AIEgens) such as tetraphenylethylene (TPE) usually possessed propeller-shaped structures and showed no or weak photoluminescence (PL) when they were molecularly dissolved in their good solvent. However, their emission was dramatically enhanced once aggregates were formed by adding poor solvent. Thus, AIE exhibited an exact opposite photophysical phenomenon to the traditional aggregation-caused quenching effect discovered by Förster in 1954. Further study revealed that the restriction of intramolecular motion (RIM) was the cause of the AIE effect. Since TPE possessed marvelous PL property in the aggregate state, it was a promising candidate to construct chiral materials with unique chiroptical properties.”

Reviewer 2:

The paper reports that easy, accurate and efficient way to monitor the conformational change of the 1,1'-binaphthyl based polymer induced by aggregation of the molecules. The authors proposed that the torsion angle of binaphthyl can be easily determined by CD spectroscopy using the relationship between $\Delta\epsilon/\Delta\lambda_{max}$ and torsion angle of the naphthalene units. The author also done the DFT calculations to interpret the experimental results. I think that the results are well organized and discussion is quite convincing; thus, the study provides an important contribution to the field of chemistry, especially structural organic chemistry field. Consequently, I would like to recommend that the manuscript should be accepted for publication in Nature Commun. Before publication, the authors should reconsider and revise the manuscript about several minor points listed below.

Many thanks for the review, such a positive feedback motivates us to further polish the manuscript. Every comment from reviewer 2 has been treated carefully. Again, there is no doubt that the comments and questions raised by the reviewer do help a lot to improve the quality of our manuscript. We hope our answers will make reviewer satisfied.

- 1) I feel that the word of “molecular dynamics” in the title gives misleading to the readers; because the proposed method can provide only an information about the static or average conformation of molecules in the solution and aggregates. I think that the word of “molecular dynamics” is associated with the dynamic change of the conformation, namely the process of the change from one conformation to the other conformation, induced by stimuli such as addition of the poor solvent. Thus, I recommend the authors to reconsider the title.*

Thanks for the valuable suggestions from the reviewer. Our original consideration to raise the title with “molecular dynamic” was that the solute experienced a continuous conformational change from water fraction (f_w) = 0 to 90%. This process was similar to adding poor solvent to the fully dissolved solution. That is the reason why we called the molecular aggregation as molecular dynamic. However, I agree with the reviewer that these two processes are different after a careful reconsideration. In our experiments, the CD spectra were recorded at the final “steady” state in mixtures with different water fractions. Thus, the monitoring was not for a whole process but was performed at a definite step. However, if the CD spectra were recorded at different time for a specific system, such monitoring may refer as “molecular dynamic”. **We finally decided to change the title from “Monitoring Molecular Dynamics Using Circular Dichroism” to “In-Situ Monitoring the Molecular Aggregation Using Circular Dichroism”.**

- 2) Because many readers of this journal is not an expert in CD spectroscopy, the author should explain the relationship between $\Delta\varepsilon$ and $[\Theta]$. They describe it briefly in p. 5 in the manuscript; however, I think it is not enough to understand their relationship.

The relationship between $\Delta\varepsilon$ and $[\Theta]$ was further introduced in the section of “Circular dichroism”. The description was as follows: “The effects of water on the CD couplet molar ellipticity $[\Theta]$ and wavelength splitting ($\Delta\lambda_{\max}$) have been intensively studied. $[\Theta]$ is a parameter to describe the molar ellipticity and is usually obtained from experiments. $\Delta\varepsilon$ is the symbol of molar circular dichroism collected from theoretical calculation. The equation of $[\Theta] \approx 3300 \Delta\varepsilon$ indicates that $[\Theta]$ is proportional to the CD couplet intensity $\Delta\varepsilon$ ”.

- 3) In fig 5, the torsion angle between two naphthalene unit is expressed with a symbol α . On the other hand, in fig 6 and main text, it is expressed with θ . It is very confusing to the readers, and the symbol must be consistent.

We are sorry for such mistake. The incorrect symbol (α) was replaced with the correct one (θ) in Figure S9.

Figure S9. Optimized structures of P-1, P-2, P-3 and P-4 in the ground state calculated by the DFT, B3LYP/6-31G(d), Gaussian 09 program. Only one repeating unit was drawn for clear presentation.

- 4) Are the aggregates of those polymers crystal? Is there any evidence if the aggregates are crystal or amorphous? If the aggregates are amorphous, the angle θ has some distribution. Is there any effect of the distribution of the θ angle on the $\Delta\varepsilon$, $[\Theta]$, and $\Delta\lambda_{\max}$?

According to our experience in the AIE research, the aggregates of polymers formed at aqueous mixtures with high water fraction (f_w) were amorphous. To verify our hypothesis, films of the polymers were prepared from their aggregates in aqueous mixture with $f_w = 90\%$. We first dropped 200 μL of aqueous suspension mixture onto quartz plate and the solvent was immediately removed by the wiping paper. This process was repeated for five times and films with fresh aggregates were prepared. This method enables the morphology of the aggregates in aqueous mixture remains even in the prepared film. After, X-ray diffraction was carried out. Figure R5 indicated that all the aggregates were amorphous, as suggested by the appearance of only a broad diffraction peak at around $2\theta = 23^\circ$ in their diffractograms.

Figure R5. XRD diffractograms of films of P-1 to P-4 prepared from their aggregate suspension in THF/water mixture (10^{-4} M with 90% water fraction).

The question on “distribution of θ ” is a good suggestion and is similar to one of the comments raised by reviewer 1. Since the aggregates are amorphous, the θ of all the polymers should be a distribution instead of a specific value. To illustrate the effect of θ distribution on $\Delta\varepsilon_{\max}$ and $\Delta\lambda_{\max}$, molecular dynamics calculation was carried out. As shown in Figure 5, the simulation results suggested that a narrow distribution of θ was

observed in “locked” P-3 with full width at half maximum (FWHM) of around 10° in both THF and water. However, the “open” P-1 showed broad θ distribution in the aggregate state, as revealed its larger FWHM of 45° . The relationship between θ and $\Delta\varepsilon_{\max}/\Delta\lambda_{\max}$ was calculated and redrawn in Figure 1. So, in each aggregate state with a specific water fraction (f_w), its CD spectrum was an average one. The strongest effect could be reflected by the plots of $[\Theta]/[\Theta]_0$ and $(\Delta\lambda_{\max} - \Delta\lambda_{\max-0})$ versus f_w in Figure 4. The annihilation of $[\Theta]$ of P-1 mainly located in the f_w range of 0 to 30%. However, the decrease of $\Delta\lambda_{\max}$ existed throughout the whole aggregation ($f_w = 0$ to 90%). Generally, this effect could be explained by the different rate change between $\Delta\varepsilon_{\max}$ and $\Delta\lambda_{\max}$ under a specific conformation (Figure 1C & D, blue line). In addition, the θ distribution also played a significant role in generating different conformation.

Figure 5. The atomistic models of P-1 and P-3 in molecular dynamics simulations. A) and E) showed single P-1 and P-3 molecule in THF solution. B) and F) demonstrated thirty P-1 and P-3 molecules aggregated in water solution. For clarity, the solvent molecules for THF and water were not shown here. C) and D) showed the relative Boltzmann populations of various conformers of P-1 in THF and water, respectively. G) and H) showed the relative Boltzmann populations of various conformers of P-3 in THF and water, respectively. FWHM: full width at half maximum.

Figure 1. A) Schematic of polarization directions of the main electronic transition moments and the torsion angle (θ) of (*R*)-1,1'-binaphthyl derivatives. B) Simulated circular dichroism spectra of (*R*)-1,1'-binaphthol with θ varying from -45 to -110° of 1° interval. C) Plot of $\Delta\epsilon_{\text{max}}/\Delta\epsilon_{\text{max-0}}$ versus θ (red line) and the first-order derivative of $d|\Delta\epsilon_{\text{max}}|/d|\theta|$ (blue line), $\Delta\epsilon_{\text{max}}$ = negative CD couplet intensity at around 220 nm and $\Delta\epsilon_{\text{max-0}} = \Delta\epsilon_{\text{max}}$ at $\theta = -45^\circ$. D) Plots of $\Delta\lambda_{\text{max}} - \Delta\lambda_{\text{max-0}}$ versus θ (red line) and the first-order derivative of $d|\Delta\lambda_{\text{max}} - \Delta\lambda_{\text{max-0}}|/d|\theta|$ (blue line). $\Delta\lambda_{\text{max}}$ = Davydov splitting width shown in Figure 1B according to the Mason model²³, $\Delta\lambda_{\text{max-0}} = \Delta\lambda_{\text{max}}$ at $\theta = -45^\circ$. Calculated by CAM-B3LYP/6-31G*, Nstates = 30, Gaussian 09 Program.

Figure 4. A) Plots of $[\Theta]/[\Theta]_0$ versus the composite of P-1 to P-4 in THF/water mixtures at $c = 10^{-4}$ M. $[\Theta]_0$ = molar ellipticity at $f_w = 0\%$. The values of $[\Theta]$ were acquired from the peaks at 280, 240, 280 and 250 nm for P-1, P-2, P-3 and P-4, respectively. B) Plots of $\Delta\lambda_s$ versus f_w , where $\Delta\lambda_s = \Delta\lambda_{\max} - \Delta\lambda_{\max-0}$ and $\Delta\lambda_{\max-0}$ = wavelength splitting at $f_w = 0\%$.

In summary, the nonlinear AACD effect in “open” polymers, was related to the broad distribution of θ .

Reviewer 3:

The modulation of the CD signal of binaphthalene when conformational changes occur is known in the literature, but here the authors use CD and photoluminescence (PL) to evaluate conformational changes upon aggregation of polymers based on (R)-1,1'-binaphthyl (BN) and tetraphenylene (TPE). Four polymers were studied (P1 to P4), depending on the location of the bond between BN and TPE, and whether BN was locked or not.

Thanks for the kind comments of reviewer. Indeed, the relationship between chiroptical properties and the conformation of 1,1'-binaphthyl has been well studied (Pu, L. 1,1'-Binaphthyl-based Chiral Materials. Imperial College Press, London (2010); Bari, L. D., Pescitelli, G. & Salvadori, P. Conformational Study of 2,2'-Homosubstituted 1,1'-Binaphthyls (BN) by Means of UV and CD Spectroscopy. *J. Am. Chem. Soc.* **121**, 7998-8004 (1999); Berova, N., Bari, L. D. & Pescitelli, G. Application of electronic circular dichroism in configurational and conformational analysis of organic compounds. *Chem. Soc. Rev.* **36**, 914-931 (2007)). From this basis, we used the CD signal of BN for in-situ monitoring the molecular aggregation. Coupled with our expertise in aggregation-induced emission (AIE), the whole package could be a pioneering tool to monitor the molecular aggregation. We hope reviewer satisfies our responses.

This paper needs major revisions.

- 1) First, the title must be changed: “Monitoring molecular dynamics using circular dichroism”. It is abusive to use the terms “molecular dynamics” of aggregation. What is observed is changes of CD and PL spectra upon adding water to a THF solution of the polymer, which causes aggregation. Then, using models relating CD and PL spectra to the torsions in the polymers, the authors try to evaluate whether the torsions change upon aggregation and in which direction (larger or smaller angles). What the authors intend to do is thus to monitor aggregation and conformational changes resulting from that aggregation, not the dynamics of the process.

This is also the same question pointed out by reviewer 2. We agree that the phrase “molecular dynamic” is inappropriate in this project after a careful study of “molecular dynamic” in other research fields. Actually, our original consideration to raise the title with “molecular dynamic” was that the solute experienced a continuous conformational change from water fraction (f_w) = 0 to 90%. This process was similar to adding poor solvent to the fully dissolved solution. That is the reason why we called the molecular aggregation as molecular dynamic. However, I agree with the reviewer that these two processes are different after a careful reconsideration. In our experiments, the CD spectra

were recorded at the final “steady” state in mixtures with different water fractions. Thus, the monitoring was not for a whole process but was performed at a definite step. However, if the CD spectra were recorded at different time for a specific system, such monitoring may refer as “molecular dynamic”. We finally decided to change the title from “Monitoring Molecular Dynamics Using Circular Dichroism” to “In-Situ Monitoring the Molecular Aggregation Using Circular Dichroism”.

- 2) In the polymers, there are two torsions that critically affect the conformations, and the authors aim to evaluate how these torsions change upon aggregation: 1) the torsion β about the bond linking tetraphenylene (TPE) and binaphthyl (BN); 2) the torsion θ within BN, β is studied by PL experiments and DFT simulations. Based on energy gaps calculated by DFT, the authors estimate that $\lambda_{(abs)}$ (PL) that is lowest in energy mainly depends on β but not θ . As $\lambda_{(abs)}$ decreases in the following order: P1 - P3 - P2 - P4 for polymers in THF, and as β (calculated by DFT) increases following the same order, the authors conclude that $\lambda_{(abs)}$ could be used to monitor β .

I think that their conclusion is not supported by the data. First, the authors did not take count in their analysis that TPE and BN are not grafted at the same positions in the four polymers, and this aspect alone can change dramatically the absorption spectra. P1 and P3 have TPE and BN grafted at the same position; P2 and P4 are grafted at another position. I think that this aspect alone explains why P1-P3 have higher $\Delta\lambda_{abs}$ than P2-P4. Then, within the group P1-P3, $\lambda_{(abs)}$ differs by 10 nm between P1 and P3, but β in the two polymers differs by only 0.7°. How can the authors thus claim that β mainly explains the shift of the absorption band in these polymers? Thus, all the analyses written from line 161 to 202, in the paragraph titled “Photophysical properties and aggregation-induced emission”, must be deeply revised if any valuable information is sought. For instance, the influence of structural parameters other than β on the absorption can be obtained by performing simulations where β is fixed in the four polymers; absorption spectra can be generated from excited states obtained from CIS calculations performed on isolated molecules using the ZINDO parameterization.

Thanks for the valuable comments. Such doubt is reasonable and convincing, and it is arbitrary to conclude that the conjugation of these four polymers is mainly depended on the torsion angle β but not θ only because the change of λ_{abs}/E_g follows the same order as β . In some senses, comparison of photophysical properties between these four polymers was inconsequential as their chemical and electronic structures were different. Following our reviewer’s suggestions, further calculations were carried out to clarify the

relationship between $\lambda_{(\text{abs})}$ and β/θ (Figure R6-7 & S10-11). Results showed that change of β contributed mainly to the shift of $\lambda_{(\text{abs})}$ in the “open” polymers (P-1 and P-2), but the decrease or increase of θ showed little influence on $\lambda_{(\text{abs})}$. However, in “locked” polymers such as P-3 and P-4, both β and θ played major roles in $\lambda_{(\text{abs})}$. To clarify this relationship, the section “**Photophysical properties and aggregation-induced emission**” in the main text was rewritten and was shown below.

Figure R6. (A and B) Simulated UV spectra of P-1 with (A) varying θ from -80° to -100° and constant β (36°), and (B) β varying from 26° to 46° and constant θ (-90°). (C) Plots of $\lambda_{\text{abs}} - (\lambda_{\text{abs}})_0$ versus θ and β . λ_{abs} : simulated absorption maximum, $(\lambda_{\text{abs}})_0$: simulated absorption maximum at $\theta = -90^\circ$ and $\beta = 36^\circ$. Semiempirical CIS/ZINDO method was applied with NStates = 10, and all these calculations were performed using Gaussian 09 Program. For the sake of clear demonstration and simplified calculation, the hexyl groups were replaced with methyl groups in P-1.

Figure S10. (A and B) Simulated UV spectra of P-2 with (A) varying θ from -70° to -90° and constant β (45°), and (B) β varying from 35° to 55° and constant θ (-80°). (C) Plots of $\lambda_{\text{abs}} - (\lambda_{\text{abs}})_0$ versus θ and β . λ_{abs} : simulated absorption maximum, $(\lambda_{\text{abs}})_0$: simulated absorption maximum at $\theta = -80^{\circ}$ and $\beta = 45^{\circ}$. Semiempirical CIS/ZINDO method was applied with NStates = 10, and all these calculations were performed using Gaussian 09 Program. For the sake of clear demonstration and simplified calculation, the hexyl groups were replaced with methyl groups in P-2.

Figure R7. (A and B) Simulated UV spectra of P-3 with (A) varying θ from -45° to -55° and constant β (37°), and (B) β varying from 32° to 42° and constant θ (-50°). (C) Plots of $\lambda_{\text{abs}} - (\lambda_{\text{abs}})_0$ versus θ and β . λ_{abs} : simulated absorption maximum, $(\lambda_{\text{abs}})_0$: simulated absorption maximum at $\theta = -50^\circ$ and $\beta = 37^\circ$. Semiempirical CIS/ZINDO method was applied with NStates = 10.

Figure S11. (A and B) Simulated UV spectra of P-4 with (A) varying θ from -46° to -56° and constant β (49°), and (B) β varying from 44° to 54° and constant θ (-51°). (C) Plots of $\lambda_{\text{abs}} - (\lambda_{\text{abs}})_0$ versus θ and β . λ_{abs} : simulated absorption maximum, $(\lambda_{\text{abs}})_0$: simulated absorption maximum at $\theta = -51^\circ$ and $\beta = 49^\circ$. Semiempirical CIS/ZINDO method was applied with NStates = 10.

Photophysical properties and aggregation-induced emission. The photophysical properties of the polymers were also studied. The UV spectra of P-1, P-2, P-3 and P-4 in THF exhibited an absorption maximum (λ_{abs}) at 347, 333, 337 and 328 nm, respectively. On the other hand, in THF/water mixture (v/v, 7/3), their emission maximum (λ_{em}) was 506, 498, 499 and 492 nm, respectively. The photophysical properties and relevant simulation results were summarized in Table 1. From the λ_{abs} and λ_{em} values, the order of conjugation was P-1 > P-3 > P-2 > P-4, which was well matched with the calculated energy gap (3.186, 3.204, 3.312 and 3.321 eV for P-1, P-3, P-2 and P-4, respectively).

Figure 7. A) UV spectra of P-1, P-2, P-3 and P-4 in THF. B) Normalized photoluminescence (PL) spectra of P-1, P-2, P-3 and P-4 in THF/water mixture (v/v, 7:3). Concentration: 10^{-5} M; λ_{ex} (nm): 355 (P-1), 345 (P-2), 345 (P-3) and 340 (P-4). Abbreviation: λ_{abs} = absorption maximum, λ_{em} = emission maximum.

Table 1. Photophysical properties and simulation results of P-1, P-2, P-3 and P-4, calculated by B3LYP/6-31G(d), Gaussian 09 program.^a

Polymer	λ_{abs} (nm)	λ_{em} (nm)	E_g (eV)	θ ($^\circ$)	β ($^\circ$)
P-1	347	506	3.186	-92.4	36.0
P-2	333	498	3.312	-78.8	44.5
P-3	337	499	3.204	-49.0	36.7
P-4	328	492	3.321	-51.2	49.5

^a Abbreviation: λ_{abs} and λ_{em} = absorption maximum and emission maximum in THF and THF/water mixture (v/v, 7:3), respectively, E_g = calculated energy gap, θ = calculated torsion angle between two naphthalene rings, β = torsion angle between the naphthalene and phenyl ring of TPE moiety. DFT calculation was carried out to optimize the ground-state conformation of these polymers in the gas phase. To simplify the calculation, only two repeating units were chosen to perform the optimization (Figure S9 and Table S2-S5).

Figure 8. (A, B, D and E) Simulated UV spectra of (A and B) P-1 and (D and E) P-3 with θ varying from (A) -80° to -100° and constant β (36°) and (D) from -45° to -55° and constant β (37°), and β varying from (B) 26° to 46° and constant θ (-90°) and (E) from 32° to 42° and constant θ (-50°). C) and F) Plots of $\lambda_{\text{abs}} - (\lambda_{\text{abs}})_0$ versus θ and β . λ_{abs} : simulated absorption maximum, $(\lambda_{\text{abs}})_0$: simulated absorption maximum wavelength of P-1 at $\theta = -90^\circ$ and $\beta = 36^\circ$; and P-3 at $\theta = -50^\circ$ and $\beta = 37^\circ$. Semiempirical CIS/ZINDO method was applied with NStates = 10 and all these calculations were performed using Gaussian 09 Program. For the purpose of a clear demonstration and simplified calculation, the hexyl groups were replaced with methyl groups in P-1.

Study on the $\lambda_{\text{abs}}/\lambda_{\text{em}}$ change could be another tool to monitor the molecular aggregation. TPE was a well-studied molecule with aggregation-induced emission characteristic. Previous studies showed that the $\lambda_{\text{abs}}/\lambda_{\text{em}}$ of TPE shifted little when its molecules aggregates. So, the relationship between λ_{abs} and β/θ was further investigated by theoretical calculation. Semiempirical CIS/ZINDO method was applied with NStates = 10 and all the calculations were performed using Gaussian 09 Program. The simulation results were shown in Figure 8 and Figure S10-11. In “open” polymer such as P-1, the λ_{abs} showed almost no change when θ decreased from -80° to -110° . However, an obvious 5 nm hypochromic shift of λ_{abs} was observed from β increased from 26° to 46° (Figure 8A-C). This indicated that the $\lambda_{\text{abs}}/\lambda_{\text{em}}$ value could be used to monitor the

change of β in P-1. However, in “locked” P-3, both β and θ contributed mainly to λ_{abs} . As shown in Figure 8D-F, decreasing θ from -45° to -55° or increasing β from 32° to 42° resulted in a 1 nm hypochromic shift of λ_{abs} . This suggested that the change of $\lambda_{\text{abs}}/\lambda_{\text{em}}$ in P-3 should be ascribed to the variation of β and θ . P-2 showed the same effect as P-1, and P-4 and P-3 also exhibited similar behavior.

Then, the PL spectra of these polymers were measured in THF/water mixtures with different f_w (Figure 9 & Figure S12-13). All the polymers showed no or quite weak emission in pure THF solution. However, their emission was tremendously enhanced upon water addition. For example, the PL intensity (I) of P-1 increased by almost 60 times at $f_w = 90\%$, demonstrative of an AIE effect (Figure 8C)²⁷⁻²⁹. Similarly, the fluorescence quantum yield (Φ_F) of the polymers also increased upon aggregate formation and their maximum solid-state Φ_F reached 60% (Figure S14 and Table S6). Except for P-2, all the polymers showed an insignificant change of λ_{em} during aggregation (Figure 9D). For P-1 and P-2, as the decrease of θ was proved to exhibit negligible impact on λ_{em} , so it could conclude that the β value remained almost unchanged from solution to aggregates in P-1. On the other hand, the 14 nm hypochromic shift of λ_{em} in P-2 suggested an increase of β in the aggregate state. In P-3 and P-4, both θ and β played important roles in the λ_{em} change. However, the λ_{em} kept almost unchanged before and after the aggregation. As suggested by the CD results, only a small increase of θ was observed in P-3 and P-4. It could be inferred that the β value also suffered small change by aggregation. In summary, among the polymers, only P-2 showed an obvious increase of β during aggregation presumably due to its crowded structure. Meanwhile, only a slight change of β was observed in the other three polymers. PL was a semiempirical method to monitor the molecular aggregation due to its environmental sensitivity. Therefore, the obtained results from PL were not as accurate as CD. One reason was that both molar ellipticity and Davydov splitting could work synergetically to monitor the conformational change. The other was that the CD measurement affected little by the external environment. All these make CD spectroscopy a promising tool to monitor the molecular aggregation.

Figure 9. PL spectra for A) P-1 and B) P-2 in THF/water mixtures with different water fractions (f_w). $c = 10^{-5}$ M. $\lambda_{ex} = 355$ nm. C) Plots of α_{AIE} versus f_w , where $\alpha_{AIE} = I/I_0$ and $I_0 =$ maximum emission intensity at $f_w = 0\%$. Inset: photographs of P-1 in THF/water mixtures with different f_w taken under UV illumination. D) Plots of $\lambda_{em} - (\lambda_{em})_0$ versus different f_w . λ_{em} : emission maximum of each plot, $(\lambda_{em})_0$: maximum emission wavelength at $f_w = 0\%$.

- 3) The ∂ torsion is studied by CD experiments coupled to the Mason's model, which calculates how the CD signal intensity ($\Delta\varepsilon$ or $[\Theta]$) and Davydov splitting ($\Delta\lambda$) evolve with θ . Based on these methods, the authors claim that upon aggregation, P1 has θ evolving from -92° (isolated chain calculated by DFT) towards -110° , as both the CD intensity and $\Delta\lambda$ simultaneously decrease in the Mason model when the angle evolves from -92° to -110° (at -110° , the signal and $\Delta\lambda$ are null), and that such evolution is also observed in CD experiments. However, when increasing the water fraction, the drops of the CD

intensity and of $\Delta\lambda$ look successive, not simultaneous. Up to 30% water, the CD signal intensity drops dramatically, then stabilizes. For $\Delta\lambda$, it is the opposite; up to 30% water, $\Delta\lambda$ drops by only ~ 1.5 nm, and after 30% water, there is another loss of 4.5 nm. How can the authors rationalize these observations? In any case, to better compare these simulation and experimental results, it should be nice to use identical graphical representation. Experimentally, $[\Theta]/[\Theta]_0$ and $\Delta\lambda-\Delta\lambda_0$ are displayed, where Θ refers to the molecule in pure THF; it should be nice to show the Mason model with $\Delta\varepsilon/\Delta\varepsilon_0$ and $\Delta\lambda-\Delta\lambda_0$, and not $\Delta\lambda$ and $\Delta\varepsilon$ only, with $\Delta\lambda_0$ and $\Delta\varepsilon_0$ corresponding to $\theta=-92^\circ$ (for P1).

That is an excellent comment. First, following the reviewer's comments, the plots in Figure 1 were redrawn. Instead of $\Delta\varepsilon$ and $\Delta\lambda$, plots of $\Delta\varepsilon_{\max}/\Delta\varepsilon_{\max-0}$ and $\Delta\lambda_{\max}/\Delta\lambda_{\max-0}$ versus θ were shown in Figure 1C and D. To make a general conclusion, $\Delta\varepsilon_{\max-0}$ and $\Delta\lambda_{\max-0}$ were obtained at $\theta = -45^\circ$ instead of -92° as the plots were drawn from -45 to -105° .

Figure 1. A) Schematic of polarization directions of the main electronic transition moments and the torsion angle (θ) of (*R*)-1,1'-binaphthyl derivatives. B) Simulated circular dichroism spectra of (*R*)-1,1'-binaphthol with θ varying from -45 to -110° of 1° interval. C) Plot of $\Delta\varepsilon_{\max}/\Delta\varepsilon_{\max-0}$ versus θ (red line) and the first-order derivative of $d(\Delta\varepsilon_{\max}/\Delta\varepsilon_{\max-0})/d\theta$ (blue line), $\Delta\varepsilon_{\max}$ = negative CD couplet intensity at around 220 nm and $\Delta\varepsilon_{\max-0} = \Delta\varepsilon_{\max}$ at $\theta = -45^\circ$. D) Plots of $\Delta\lambda_{\max}/\Delta\lambda_{\max-0}$ versus θ (red line) and the first-order derivative of $d(\Delta\lambda_{\max}/\Delta\lambda_{\max-0})/d\theta$ (blue line).

$d|\Delta\lambda_{\max}|/d|\theta|$ (blue line). $\Delta\lambda_{\max}$ = Davydov splitting width shown in Figure 1B according to the Mason model²³, $\Delta\lambda_{\max-0} = \Delta\lambda_{\max}$ at $\theta = -45^\circ$. Calculated by CAM-B3LYP/6-31G*, Nstates = 30, Gaussian 09 Program.

Experimentally, the annihilation of $[\Theta]$ was observed mainly located at f_w range of 0-30%. However, the $\Delta\lambda_{\max}$ dropped at f_w from 0 to 90%. This is an interesting phenomenon and its generating mechanism could be explained by Figure 1C & D. The blue line in Figure 1C showed the plot of $d|\Delta\epsilon_{\max}|/d|\theta|$ versus θ , from which the rate of $\Delta\epsilon_{\max}/\Delta\epsilon_{\max-0}$ change at different θ could be deduced. Similarly, the rate of $(\Delta\lambda_{\max} - \Delta\lambda_{\max-0})$ change could be obtained from the plot of $d|\Delta\lambda_{\max}|/d|\theta|$ (blue line) in Figure 1D. For P-1 and P-2, the optimized θ in the gas phase was -92° and -79° , respectively. These conformers showed the fastest annihilation $\Delta\epsilon_{\max}$ at a rate of -0.4. However, their $\Delta\lambda_{\max}$ decreased in a much slower rate of 0.05. As we proposed, the θ got close to -110° with an increase of water fraction. So, it was reasonable that the annihilation of $[\Theta]$ occurred mainly at the beginning of aggregation as the change of θ during this process resulted in a dramatic decrease of $\Delta\epsilon_{\max}$. After that, the annihilation became weaker. However, the initial rate of $\Delta\lambda_{\max}$ decrease was quite small, that is the reason why no tremendous change of $(\Delta\lambda_{\max} - \Delta\lambda_{\max-0})$ was observed before $f_w = 30\%$. Afterwards, a smooth decrease in $\Delta\lambda_{\max}$ was observed. In summary, we proposed that the different dependence of $[\Theta]/[\Theta]_0$ and $(\Delta\lambda_{\max} - \Delta\lambda_{\max-0})$ on f_w was due to the distinct rate of $\Delta\epsilon_{\max}$ and $\Delta\lambda_{\max}$ change at each θ .

- 4) *In addition, I think there are errors in the graphics of Figure 4, as these graphics are not totally consistent with the graphics of Figure 3. For P1, the dots at 80 and 90% water must probably be permuted, as the color coding in Figure 3 indicates that the strongest attenuation of intensity is for 90% water; for P2, at 10% water, the signal is attenuated by 10% while in Figure 3 the signal has the same intensity as for 0% of water; also for P2, for 60% water, the signal is attenuated by 80%, while in Figure 3, the attenuation looks more close to 50%.*

We really feel sorry for such a careless mistake. The original data were carefully checked to figure out the reason for these mistakes and ensure the reliability of all our data. i) All the CD spectra presented in Figure 3 were consistent with the original data. When we plotted the relationship between $[\Theta]/[\Theta]_0$ and water fraction (f_w) for P-1, the data points at 80 and 90% were permuted. The correct curve was replotted in Figure 4A. ii) Analysis of P-2 has spent us a long time as its CD signal was apparently different from the others. We previously attributed it to its large twist angle $|\theta|$ (may be larger than 110°). Then, the

data points of $[\Theta]$ were selected at around 270 but not 240 nm. That was the reason why the plot of $[\Theta]/[\Theta]_0$ versus f_w of P-2 in Figure 4A did not match with Figure 3B. After that, the θ of P-2 was optimized at -79° in gas phase and the crystal of its monomer showed a twist angle of -85° which was less negative than P-1. Then, we realized that the CD signal of P-2 was not opposite to the others polymers, but the positive peak of Davydov splitting was located at below 220 nm which was even shorter than the cut-off wavelength of THF. Finally, the data points at around 240 nm were picked out to plot the curve of $[\Theta]/[\Theta]_0$ versus f_w for P-2. Unfortunately, we did not show the correct plot for P-2 in Figure 4A, sorry again for our careless mistake. The right curve for P-2 was replotted in Figure 4A (see below).

Figure 3. CD spectra of A) P-1, B) P-2, C) P-3 and D) P-4 in THF/water mixtures with different water fraction (f_w). $c = 10^{-4}$ M.

Figure 4. A) Plots of $[\Theta]/[\Theta]_0$ versus the composite of P-1 to P-4 in THF/water mixtures at $c = 10^{-4}$ M. $[\Theta]_0$ = molar ellipticity at $f_w = 0\%$. The values of $[\Theta]$ were acquired from the peaks at 280, 240, 280 and 250 nm for P-1, P-2, P-3 and P-4, respectively. B) Plots of $\Delta\lambda_s$ versus f_w , where $\Delta\lambda_s = \Delta\lambda_{\max} - \Delta\lambda_{\max-0}$ and $\Delta\lambda_{\max-0}$ = wavelength splitting at $f_w = 0\%$.

Reviewer #2 (Remarks to the Author):

I have reviewed the paper entitled "In-situ monitoring the molecular aggregation using circular dichroism" by Prof. Tang et al. again carefully. I feel that the manuscript has been revised well in the light of my comments. Consequently, I recommend that the manuscript of this version should be accepted for publication in Nature Commun.

Reviewer #3 (Remarks to the Author):

The authors replied appropriately to the revisions proposed and brought new information to their paper, so the quality has improved. The paper thus now needs only a few minor revisions before acceptance:

- 1) Line 172 : the sign « - » is missing in front of « 102° ».
- 2) In the discussion about Photophysical properties and aggregation-induced emission, it is concluded that β changes (or not) during aggregation, depending on the polymer (a strong change expected for P2, a weak one for the other polymers). Because the authors did MD calculations on P1 and P3, they have access to the population distributions of β for the polymers in solution and in aggregated state. Thus, it should be a nice confirmation, and quick to do, to show these populations distributions, as the authors did for θ in Figure 5.

Then, two comments:

- 1) I would like to come back to the revision 3) that I suggested in my first review. I then wrote:
when increasing the water fraction, the drops of the CD intensity and of $\Delta\lambda$ look successive, not simultaneous. Up to 30% water, the CD signal intensity drops dramatically, then stabilizes. For $\Delta\lambda$, it is the opposite; up to 30% water, $\Delta\lambda$ drops by only ~ 1.5 nm, and after 30% water, there is another loss of 4.5 nm. How can the authors rationalize these observations?

The authors then replied:

Experimentally, the annihilation of $[\Theta]$ was observed mainly located at fw range of 0-30%.

However, the $\Delta\lambda_{\max}$ dropped at fw from 0 to 90%. This is an interesting phenomenon and its generating mechanism could be explained by Figure 1C & D. The blue line in Figure 1C showed the plot of $d|\Delta\epsilon_{\max}|/d|\theta|$ versus θ , from which the rate of $\Delta\epsilon_{\max}/\Delta\epsilon_{\max-0}$ change at different θ could be deduced. Similarly, the rate of $(\Delta\lambda_{\max} - \Delta\lambda_{\max-0})$ change could be

obtained from the plot of $d|\Delta\lambda_{\max}|/d|\theta|$ (blue line) in Figure 1D. For P-1 and P-2, the optimized θ in the gas phase was -92° and -79° , respectively. These conformers showed the fastest annihilation $\Delta\epsilon_{\max}$ at a rate of -0.4 . However, their $\Delta\lambda_{\max}$ decreased in a much slower rate of 0.05 .

I do not agree when the authors compare absolute values of the rates of changes of $\Delta\epsilon_{\max}$ and of $\Delta\lambda_{\max}$. The physical quantities and units are different, and thus cannot be compared. While the authors say that the rate of change of $\Delta\lambda_{\max}$ is small for θ in the gas phase at -92° and -79° , it is even smaller when θ is close to -110° , i.e. it follows the same behavior as the rate of change of $\Delta\epsilon_{\max}$ with θ . Hence, the behavior of $\Delta\lambda_{\max}$ should be similar to the one of $\Delta\epsilon_{\max}$. I guess that the explanation about the discrepancies between the behaviours of $\Delta\lambda_{\max}$ and $\Delta\epsilon_{\max}$ is to be searched in the complexity of the reality. It was suggested that θ evolves from -92 or -79° to -110° during aggregation, but it is a simple description of the reality. Instead, the MD calculations have shown that the population distribution of θ evolves from monomodal to bimodal. Maybe it is in that direction that we need to find an explanation. As expected, there are less conformations at -90° and more towards -110° , but – though it is difficult to say from the graphics – it seems that there are also more conformations towards -60° . If we multiply the population distribution of Figure 5 by the red curves of Figure 1, could we get a rough idea of $\Delta\epsilon_{\max}$ and $\Delta\lambda_{\max}$ before and after aggregation?

2) I wonder how the authors got the blue curves in Figure 1; they are not smooth, while they are supposed to be the derivative of the red fit curves, right? What function has been used to fit the red dots?

Response Letter to the Reviewers

Dear reviewers,

On behalf of all the contributing authors, we would like to thank all the valuable comments from you, which help us a lot to improve the quality of the manuscript. The reviewers' comments are laid out below in italicized and underlined font and specific concerns have been numbered. Our response is given in normal font and the changes/additions are given in red color in the revised manuscript.

Reviewer 2:

I have reviewed the paper entitled “In-situ monitoring the molecular aggregation using circular dichroism” by Prof. Tang et al. again carefully. I feel that the manuscript has been revised well in the light of my comments. Consequently, I recommend that the manuscript of this version should be accepted for publication in Nature Commun.

We do appreciate such a positive feedback from reviewer and thanks for spending your valuable time to revise the manuscript again. We hope our revised version is qualified to be published in *Nat. Commun.*

Reviewer 3:

The authors replied appropriately to the revisions proposed and brought new information to their paper, so the quality has improved. The paper thus now needs only a few minor revisions before acceptance:

There is no doubt that the reviewer is a real scientist with rigorous scientific attitude and excellent scientific literacy. We are quite sure that he/she has carefully revised our MS word by word and digested almost all the information, which could be reflected by his/her comments both in the first and second-round. Obviously, the quality of our manuscript is improved a lot under the guidance of these comments. We do appreciate the helpful comments from the reviewer and hope our response will make him/her satisfied.

1) Line 172 : the sign « - » is missing in front of « 102° ».

We are really sorry for this mistake. The symbol of “-” has already been added in the main text and highlighted with red color.

2) In the discussion about Photophysical properties and aggregation-induced emission, it is concluded that β changes (or not) during aggregation, depending on the polymer (a strong change expected for P2, a weak one for the other polymers). Because the authors did MD calculations on P1 and P3, they have access to the population distributions of β for the polymers in solution and in aggregated state. Thus, it should be a nice confirmation, and quick to do, to show these populations distributions, as the authors did for θ in Figure 5.

Thanks for the good suggestions. The distribution of β for P-1 and P-3 obtained from the MD simulation has been extracted and the corresponding results were shown in Figure S20 - 23. Unexpectedly, the dihedral angle β was not fixed. For example, in P-1, there was an interchange between the conformers with $\beta \approx 30^\circ$ and -30° both in THF and water environment, which means the rotation of the relevant phenyls rings happened all the time. P-3 almost had the same effect with P-1 on the performance of β . Furthermore, there was extra two kind of conformers in P-3 with $\beta \approx 150^\circ$ and -150° . However, from the point of view of electronic conjugation, the energy gap almost showed no difference among $\beta = 30^\circ, -30^\circ, 150^\circ$ and -150° . Then, in order to verify the reliability of photoluminescence (PL) results, all the angles of β were transformed and centralized in the range of $0 - 90^\circ$. In this case, the angle with $\beta = 30^\circ, -30^\circ, 150^\circ$ and -150° could be classified into $\beta = 30^\circ$. The processed data were exhibited in Figure 10.

Figure S20. Distribution of dihedral angle β of P-1 in THF solution. A) to D) Representing four independent MD trajectories.

Figure S21. Distribution of dihedral angle β of P-1 aggregate in water solution. A) to F) Representing six independent MD trajectories.

Figure S22. Distribution of dihedral angle β of P-3 in THF solution. A) to D) Representing four independent MD trajectories.

Figure S23. Distribution of dihedral angle β of P-3 aggregate in water solution. A) to F) Representing six independent MD trajectories.

As shown in Figure 10, from THF to water, the distribution of processed β has not shown an obvious change both in P-1 and P-3. Meanwhile, their most probable angle is also similar. For example, P-1 and P-3 exhibited the highest probability of β at 28° and 32° , respectively, both in THF and water, which was consistent with the PL results apparently. Based on the above analysis, it could be further concluded that the PL method cannot accurately monitor the conformational change in the process of aggregation, which is due to the insufficient information provided by the PL spectra.

Figure 10. A) and B) showed the relative Boltzmann populations of dihedral angle β in various conformers of P-1 in THF and water, respectively. C) and D) showed the relative Boltzmann populations of dihedral angle β in various conformers of P-3 in THF and water, respectively.

We added this part into the main text to further demonstrate the advantage of CD than PL in terms of monitoring the conformational change.

Among these polymers, it seems that only P-2 showed an obvious increase of β during aggregation presumably due to its crowded structure, and only a slight change of β existed in the other three polymers. In order to verify this conclusion, the distribution of β for P-1 and P-3 obtained from the MD simulation was extracted and the corresponding results were shown in Figure S20 -23. Unexpectedly, the dihedral angle β was not fixed both in THF and water. There was an interchange between the conformers with $\beta \approx 30^\circ$, -30° , 150° and -150° , which indicated the phenyl ring rotation was active in these polymers but with several probable conformers. From the point of view of electronic conjugation, the energy gap almost showed no difference among $\beta = 30^\circ$, -30° , 150° and -150° . Then all the angles of β were transformed and centralized in the range of $0 - 90^\circ$. In this case, the angle with $\beta = 30^\circ$, -30° , 150° and -150° could be classified into $\beta = 30^\circ$. The processed data were exhibited in Figure 10. From THF to water, the distribution of processed β has not shown an obvious change both in P-1 and P-3. Meanwhile, their most probable angle is also similar. For example, P-1 and P-3 showed the highest probability of β at 28° and 32° , respectively, both in THF and water, which was consistent with the PL results apparently. Based on the above analysis, it could be further concluded that the PL method cannot accurately monitor the conformational change in the process of aggregation, which is due to the insufficient information provided by the PL spectra. On the contrary, in CD spectra, both molar ellipticity and Davydov splitting could work synergetically to monitor the conformational change, which made CD spectroscopy a promising tool to monitor the molecular aggregation.

Response to the following two comments:

1) I would like to come back to the revision 3) that I suggested in my first review. I then wrote: “when increasing the water fraction, the drops of the CD intensity and of $\Delta\lambda$ look successive, not simultaneous. Up to 30% water, the CD signal intensity drops dramatically, then stabilizes. For $\Delta\lambda$, it is the opposite; up to 30% water, $\Delta\lambda$ drops by only ~ 1.5 nm, and after 30% water, there is another loss of 4.5 nm. How can the authors rationalize these observations?” The authors then replied: “Experimentally, the annihilation of $[\Theta]$ was observed mainly located at f_w range of 0-30%. However, the $\Delta\lambda_{max}$ dropped at f_w from 0 to 90%. This is an interesting phenomenon and its generating mechanism could be explained by Figure 1C & D. The blue line in Figure 1C showed the plot of $d_{|\Delta\epsilon_{max}|}/d_{|\theta|}$ versus θ , from which the rate of $\Delta\epsilon_{max}/\Delta\epsilon_{max-0}$

change at different θ could be deduced. Similarly, the rate of $(\Delta\lambda_{\max} - \Delta\lambda_{\max-0})$ change could be obtained from the plot of $d|\Delta\lambda_{\max}|/d|\theta|$ (blue line) in Figure 1D. For P-1 and P-2, the optimized θ in the gas phase was -92° and -79° , respectively. These conformers showed the fastest annihilation $\Delta\varepsilon_{\max}$ at a rate of -0.4 . However, their $\Delta\lambda_{\max}$ decreased in a much slower rate of 0.05 .”

I do not agree when the authors compare absolute values of the rates of changes of $\Delta\varepsilon_{\max}$ and of $\Delta\lambda_{\max}$. The physical quantities and units are different, and thus cannot be compared. While the authors say that the rate of change of $\Delta\lambda_{\max}$ is small for θ in the gas phase at -92° and -79° , it is even smaller when θ is close to -110° , i.e. it follows the same behavior as the rate of change of $\Delta\varepsilon_{\max}$ with θ . Hence, the behavior of $\Delta\lambda_{\max}$ should be similar to the one of $\Delta\varepsilon_{\max}$. I guess that the explanation about the discrepancies between the behaviors of $\Delta\lambda_{\max}$ and $\Delta\varepsilon_{\max}$ is to be searched in the complexity of the reality. It was suggested that θ evolves from -92 or -79° to -110° during aggregation, but it is a simple description of the reality. Instead, the MD calculations have shown that the population distribution of θ evolves from monomodal to bimodal. Maybe it is in that direction that we need to find an explanation. As expected, there are less conformations at -90° and more towards -110° , but – though it is difficult to say from the graphics – it seems that there are also more conformations towards -60° . If we multiply the population distribution of Figure 5 by the red curves of Figure 1, could we get a rough idea of $\Delta\varepsilon_{\max}$ and $\Delta\lambda_{\max}$ before and after aggregation?

Thanks for the valuable comments from reviewer. The different behaviors of $[\Theta]/[\Theta]_0$ and $\Delta\lambda_s$ within the process of aggregation is really interesting (Figure 4). We tried to use the different change rate of $\Delta\varepsilon_{\max}$ and $\Delta\lambda_{\max}$ versus θ to explain this abnormal phenomenon. Indeed, as the reviewer argued, the different physical quantities and units of these two parameters made this comparison unreasonable. After much deliberation, we agree that the change rate may not be the real reason as the change rate of $\Delta\varepsilon_{\max}/\Delta\varepsilon_{\max-0}$ and $\Delta\lambda_{\max} - \Delta\lambda_{\max-0}$ both are decreased from -92° or -79° to -110° , and comparison of the absolute value between $d|\Delta\varepsilon_{\max}|/d|\theta|$ and $d|\Delta\lambda_{\max}|/d|\theta|$ is meaningless.

As reviewer suggested, the MD simulation indicated that the θ was not evolving from one angle to another one when the media was changed from THF to water. The real situation was the change of its distribution (as shown in Figure 5). So the performance of CD signal was controlled by a comprehensive effect of different conformers. In any case, we have multiplied the population distribution of Figure 5 by the red curves of Figure 1. Table R1 showed that, from THF to water, P-1 exhibited a relative big decrease in $\Delta\varepsilon_{\max}$ and $\Delta\lambda_{\max}$ but P-3 almost kept no change, which was consistent with the experimental results. However, as only the initial and final state could be read from Table R1, no more information could be obtained for the intermediate state, for example, $f_w = 30\%$.

Table R1. Calculated $\Delta\varepsilon_{\max}$ and $\Delta\lambda_{\max}$ for P-1 and P-3 in THF and water.

THF		Water	
$\Delta\varepsilon_{\max}$	$\Delta\lambda_{\max}$	$\Delta\varepsilon_{\max}$	$\Delta\lambda_{\max}$

P-1	-6.05173	18.77667	-5.81068	17.71473
P-3	-9.30729	24.52628	-9.23432	24.46487

Whereas, when the CD spectra of (*R*)-1,1'-binaphthol were simulated with the θ in the range of -20 to -130°, we found its CD spectra were more complicated than the information given in Figure 1C & D. Especially when the θ was located around 100°, which showed a signal inversion. When the θ was bigger than 100°, the Davydov splitting width ($\Delta\lambda_{\max}$) was difficult to be extracted out as the shape of the CD spectra have already changed. In other words, the experimental spectrum was the superposition of many different spectra from each conformer. Then, the observed $\Delta\epsilon_{\max}$ and $\Delta\lambda_{\max}$ from the measurement were the statistic result. Simply multiply the population distribution by the $\Delta\epsilon_{\max}$ or $\Delta\lambda_{\max}$ may not reflect the real situation as the wavelength corresponding to the $\Delta\epsilon_{\max}$ was changed with θ . In summary, the desynchrony of the change of $\Delta\epsilon_{\max}$ or $\Delta\lambda_{\max}$ should be ascribed to the multi-dispersive characteristic of θ both in THF and water. Then, the obtained CD spectra at different f_w reflected the population change of θ from solution to aggregation.

Figure R1. Simulated circular dichroism spectra of (*R*)-1,1'-binaphthol with the different dihedral angle of θ .

2) I wonder how the authors got the blue curves in Figure 1; they are not smooth, while they are supposed to be the derivative of the red fit curves, right? What function has been used to fit the red dots?

Yes, the blue curves in Figure 1C & D were the first-order derivative of the corresponding red fit curves. Meanwhile, the red dots in Figure 1C & D were smoothed with Savitzky–Golay method. We added this information in the main text.

Reviewer #3 (Remarks to the Author):

I have reviewed the paper "In-Situ Monitoring the Molecular Aggregation Using Circular Dichroism" and think it is ready for publication after the improvements made by the authors (correct the misprint in Figure 5, where α is θ , and check that A and B in Figure 10 are the correct pictures, as well as the pictures in Figure S20 and S21; they are the same).

Response Letter to the Reviewers

Reviewer 3:

I have reviewed the paper "In-Situ Monitoring the Molecular Aggregation Using Circular Dichroism" and think it is ready for publication after the improvements made by the authors (correct the misprint in Figure 5, where α is θ , and check that A and B in Figure 10 are the correct pictures, as well as the pictures in Figure S20 and S21; they are the same).

We sincerely thank the positive feedback from the reviewer and appreciate he/she agrees to publish our revised version in *Nature Communication*. For the improvements, the symbol of “ α ” in Figure 5 caption has been corrected to “ θ ”. Meanwhile, we apologize that some incorrect figures are put in Figure 10 A &B, Figure S20 and S21, these mistakes have been corrected in the main text and ESI. Our conclusion in the main text has not been affected by these mistakes. Once again, we feel sorry for our carelessness and thank reviewer for spending their valuable time to revise our manuscript time after time. The quality of our manuscript is improved a lot under the guidance of these comments.